# Hot Spots Drift in Synchronous and Asynchronous Polars: Results of Three-Dimensional Numerical Simulation

**Dmitry Bisikalo, Andrey Sobolev * and Andrey Zhilkin** 

Institute of Astronomy of the Russian Academy of Sciences, 48 Pyatnitskaya St., 119017 Moscow, Russia; bisikalo@inasan.ru (D.B.); zhilkin@inasan.ru (A.Z.)
* Correspondence: asobolev@inasan.ru

**Abstract:** In this paper, the characteristics of hot spots on an accretor surface are investigated for two types of polars: the eclipsing synchronous polar V808 Aur and the non-eclipsing asynchronous polar CD Ind in configuration of an offset and non-offset magnetic dipole. The drift of hot spots is analyzed based on the results of numerical calculations and maps of the temperature distribution over the accretor surface. It is shown that a noticeable displacement of the spots is determined by the ratio of ballistic and magnetic parts of the jet trajectory. In the synchronous polar, the dominant influence on the drift of hot spots is exerted by variations in the mass transfer rate, which entail a change in the ballistic part of the trajectory. It was found that when the mass transfer rate changes within the range of $10^{-10} M_\odot$/year to $10^{-7} M_\odot$/year, the displacement of the hot spot in latitude and longitude can reach $30°$. In the asynchronous polar, a change in the position of hot spots is mainly defined by the properties of the white dwarf magnetosphere, and the displacement of hot spots in latitude and longitude can reach $20°$.

**Keywords:** close binary star; polar; MHD; flow structure; donor; accretor; hot spot; temperature map

---





## 1. Introduction

Polars [1] are close binary stars; their optical and infrared radiation is characterized by a significant degree of polarization that is reflected in their name. As a rule, they consist of a white dwarf (accretor, the primary component) and a low-mass main sequence star (donor, the secondary component). During the evolution of a binary system, the donor overflows its Roche lobe and begins to lose matter from its atmosphere through the internal Lagrange point, which then has an accretion to the primary component [2–5]. Polars have a significant magnetic field (as a rule, the magnitude of magnetic field induction of a white dwarf exceeds 10 MGs); therefore, instead of an accretion disk, collimated streams of matter are formed in the system, moving along the field lines and accreting in the vicinity of the magnetic poles of the white dwarf.

In addition to influencing the structure of the accretion flow, a strong magnetic field in the polars contributes to the synchronization of the proper rotation periods of its components and the orbital period of the binary system [6,7]. However, flare activity on the surface of a white dwarf can lead to some asynchronism of the accretor's own rotation (on the order of 1–2%) [8]. In this regard, two classes of polars are distinguished—synchronous and asynchronous (BY Cam-type systems). Currently, about 100 polars with an orbital period of 80 minutes to 8 hours are known, and only 4 of them are asynchronous: V1500 Cyg [9], 1432 Aql [10], BY Cam [11], and CD Ind [12].

In the structure of accretion flow for any type of polar, it is possible to distinguish ballistic and magnetic parts. The ballistic section is bounded by the area between the inner Lagrange point $L_1$, and the boundary is defined by the Alfven radius of the accretor. In this area, a movement of matter is determined mainly by the forces of gravity and inertia [13]. At the same time, the matter pressure does not play a significant role, due to the supersonic nature of the flow. The length of the ballistic section of accretion flow can vary depending

---

on values of the mass transfer rate and magnetic field induction: with an increase in the mass transfer rate and a decrease in the field induction, the ballistic trajectory lengthens. On the magnetic part of the trajectory (within the magnetosphere), electromagnetic forces become dominant, and therefore, the accreting matter moves along the magnetic force lines [14].

The lengths ratio of both the ballistic and magnetic parts of the jet trajectory determines the position of a hot spot relatively to the magnetic pole. At low values of the mass transfer rate, the matter accretion is mainly specified by the properties of magnetosphere and occurs in the vicinity of the magnetic poles region. With an increase in the mass transfer rate and a corresponding elongation of the ballistic trajectory, the accreting flow deviates more strongly from the polar region since it turns out to be on the magnetic line leaving the accretor at some distance from the pole. In addition, the jet deflection is facilitated by action of the Coriolis force: the more force, the more deviation in the direction opposite to the rotation of a binary system.

Change in the position of hot spots relative to the magnetic poles can occur both in longitude and in latitude of the primary component [15]. In the case of a synchronous polar and under the assumption of a dipole configuration of the magnetic field in which the center of the dipole coincides with the center of a white dwarf (a non-offset dipole), the position of a hot spot is uniquely determined by the mass transfer rate. In this case, both the latitude and longitude of the energy release zones will vary since they are located symmetrically relative to the orbital plane of a binary system. In the case of an asynchronous polar and under the assumption of the fixed mass transfer rate, the rotation of the dipole axis in time begins to play a significant role in the spots drift—a change in its orientation during the beat period relative to the donor. The flow pattern becomes even more complicated for the case of an offset dipole when its center lies above or below the orbital plane of a polar. This paper is devoted to investigation of the drift spots parameters for the cases of synchronous and asynchronous polars.

The paper is organized as follows. The second section describes the numerical model we use. The third section presents the results of numerical calculations. In conclusion, the main inferences of the paper are briefly discussed.

## 2. Numerical Model

The calculation method for studying the drift of hot spots in the polars includes two stages:

(1) Conducting a three-dimensional numerical MHD simulation of the flow structure in a binary system;

(2) Calculating the temperature distribution over the accretor surface.

At the first stage, when calculating the flow structure, two versions of the numerical model are used—stationary and non-stationary. The first one involves obtaining a sequence of solutions in which the position of the accretor regarding the donor is fixed for a given period of time. This time interval is chosen based on the condition for the solution to enter a quasi-stationary mode, and, as a rule, is at least 1–2 orbital periods.

In the case of an asynchronous polar, using the stationary model allows us to obtain a consistent set of fixed solutions corresponding to different values of the beat period phases. It is obvious that such an approach can only provide a general picture of changes in the flow structure over a sufficiently long beat period, but it is not suitable for studying the rapid dynamics of the flow on a small time scale. In the non-stationary model, the proper rotation of the accretor is set to be continuous regardless of the selected time scale for the output of numerical solutions, which corresponds to the real system as much as possible and allows us to study rapidly occurring changes in its flow structure more correctly.

To describe the flow structure in polars, we have developed a self-consistent three-dimensional numerical model [4,5,16]. This model is based on the equations system of modified magnetic hydrodynamics, which describes astrophysical flows under conditions of a strong external magnetic field. We have successfully applied this approach to modeling the flow structure for both polars and intermediate polars [5,16–28].

The numerical simulation was carried out in a non-inertial reference frame rotating together with a binary system having an angular velocity of $\Omega = 2\pi/P_{\text{orb}}$ around its mass center, where $P_{\text{orb}}$ is the orbital period of a polar. For an asynchronous polar, three parameters are used to characterize its rotation: $P_{\text{beat}}$—the beat period; $P_{\text{spin}}$—the proper period of asynchronous rotation of the accretor; and $P_{\text{orb}}$. They are related by the following expression:

$$\frac{1}{P_{\text{beat}}} = \frac{1}{P_{\text{spin}}} - \frac{1}{P_{\text{orb}}}. \tag{1}$$

The field of forces acting on matter in such a reference frame is determined by the Roche potential [4]:

$$\Phi = -\frac{GM_{\text{a}}}{|\boldsymbol{r} - \boldsymbol{r}_{\text{a}}|} - \frac{GM_{\text{d}}}{|\boldsymbol{r} - \boldsymbol{r}_{\text{d}}|} - \frac{1}{2}[\boldsymbol{\Omega} \times (\boldsymbol{r} - \boldsymbol{r}_{\text{c}})]^2, \tag{2}$$

where $G$ is the gravitational constant; $\boldsymbol{r}_{\text{a}}$, $\boldsymbol{r}_{\text{d}}$ and $\boldsymbol{r}_{\text{c}}$ are the radius vectors that determine the position of the accretor center, donor center and mass center of the system, respectively; $\mathbf{r}$ is the radius vector of an arbitrary point of the system; $M_{\text{a}}$ and $M_{\text{d}}$ are the accretor and donor mass, respectively; and $\boldsymbol{\Omega} = (0, 0, \Omega)$ is the vector of the angular velocity of a binary system rotation. The first and the second terms in expression (2) describe both the accretor and donor gravitational potential. The last term describes the centrifugal potential relative to the mass center.

The selected reference frame uses the Cartesian coordinate system $(x, y, z)$, which is schematically represented in Figure 1.

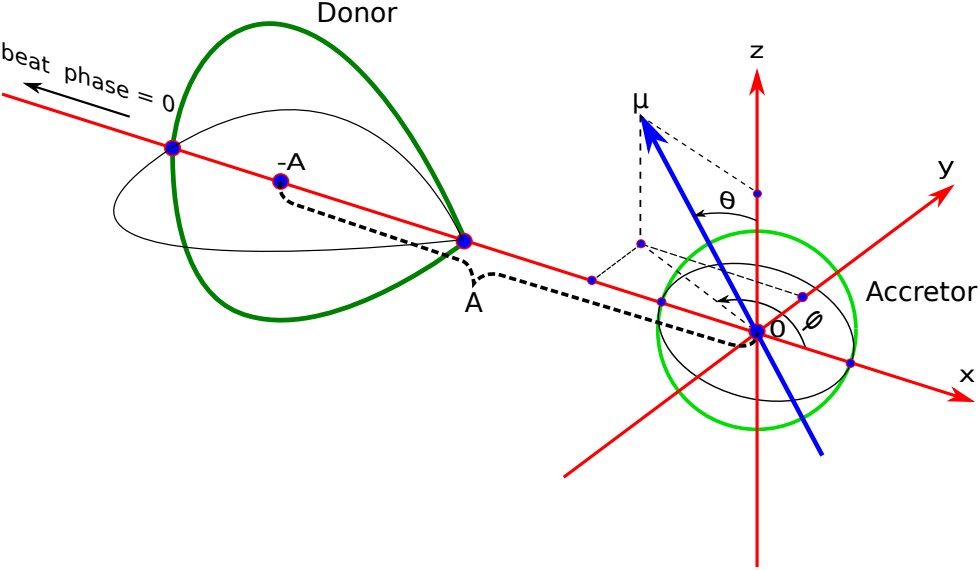

**Figure 1.** The Cartesian coordinate system used in the numerical model.

A magnitude of the magnetic induction at an arbitrary point of a binary system can be described by the following expression:

$$\mathbf{B}_* = \frac{\mu}{R^3}[3(\mathbf{d} \cdot \mathbf{n})\mathbf{n} - \mathbf{d}], \tag{3}$$

where $\mu = B_{\text{a}} R_{\text{a}}^3 / 2$ is the accretor magnetic moment; $B_{\text{a}}$ is the characteristic value of the magnetic field induction on the surface of a white dwarf, obtained from observations; $R_{\text{a}}$ is the accretor radius; $R = |\mathbf{R}|$ is the distance from the center of the magnetic dipole to the observation point of field; and $\mathbf{n} = \mathbf{R}/R$ is the unit vector of normal to the sphere of radius

$R$, raised at the field observation point. The unit vector **d** defines the symmetry axis of the dipole. The components of vector **d** in the Cartesian coordinate system can be written as:

$$d_x = \sin\theta\cos\phi, \quad d_y = \sin\theta\sin\phi, \quad d_z = \cos\theta, \tag{4}$$

where the angles $\theta$, $\phi$ determine the spacial orientation of magnetic axis (see Figure 1). In the case of a synchronous polar, the angle $\phi$ is constant in time and is equal to the initial value corresponding to the zero orbital phase. For an asynchronous polar, this angle already depends on time and increases during the orbital by the orbital period transition according to the following law:

$$\phi = \phi_0 + \Omega_{\text{beat}}t, \tag{5}$$

where $\Omega_{\text{beat}} = 2\pi/P_{\text{beat}}$, $t = P_{\text{orb}}N$ and $N$ are the quantity of orbital periods from the zero phase of the beat period. The angle $\theta$ in the systems under consideration always remains constant.

Note that the magnetic field $\mathbf{B}_*$, given by Formula (3), is irrotational, $\nabla \times \mathbf{B}_* = 0$. The variability or constancy in time of the azimuthal angle $\phi$ for the polars under consideration leads to the corresponding character of magnetic field. So, for a synchronous polar, it will be stationary, $\partial\mathbf{B}_*/\partial t = 0$, and for an asynchronous, non-stationary, $\partial\mathbf{B}_*/\partial t \neq 0$. The irrotationality of the magnetic field allows it to be partially excluded from the corresponding equations describing the structure of MHD flow [4,29–31]. This technique in the numerical model is convenient to use in order to avoid the accumulation of errors during operations with large numbers in the calculation process. To do this, the total magnetic field **B** can be represented as a superposition of the accretor field $\mathbf{B}_*$ and the field **b** induced by electric currents in the accretion jet and envelope of a binary system: $\mathbf{B} = \mathbf{B}_* + \mathbf{b}$.

In the case of asynchronous rotation of the accretor, the corresponding change in these field components over time is described by the following equations [4,5]:

$$\frac{\partial\mathbf{B}}{\partial t} = \frac{\partial\mathbf{b}}{\partial t} + \frac{\partial\mathbf{B}_*}{\partial t}, \tag{6}$$

$$\frac{\partial\mathbf{B}_*}{\partial t} = \nabla \times (\mathbf{v}_* \times \mathbf{B}_*). \tag{7}$$

Here $\mathbf{v}_* = \boldsymbol{\Omega}_{\text{beat}} \times (\mathbf{r} - \mathbf{r}_{\text{a}})$ is the magnetic field line velocity, vector $\boldsymbol{\Omega}_{\text{beat}} = (0, 0, \Omega_{\text{beat}})$. The change of the component **b** in time is expressed by the induction Equation (10).

The following system of MHD equations was used for simulation of the flow structure:

$$\frac{\partial\rho}{\partial t} + \nabla \cdot (\rho\mathbf{v}) = 0, \tag{8}$$

$$\frac{\partial\mathbf{v}}{\partial t} + (\mathbf{v} \cdot \nabla)\mathbf{v} = -\frac{\nabla P}{\rho} - \frac{\mathbf{b} \times (\nabla \times \mathbf{b})}{4\pi\rho} - \nabla\Phi + 2(\mathbf{v} \times \boldsymbol{\Omega}) - \frac{(\mathbf{v} - \mathbf{v}_*)_\perp}{t_w}, \tag{9}$$

$$\frac{\partial\mathbf{b}}{\partial t} = \nabla \times [\mathbf{v} \times \mathbf{b} + (\mathbf{v} - \mathbf{v}_*) \times \mathbf{B}_* - \eta_w(\nabla \times \mathbf{b})], \tag{10}$$

$$\rho\left[\frac{\partial\varepsilon}{\partial t} + (\mathbf{v} \cdot \nabla)\varepsilon\right] = -P(\nabla \cdot \mathbf{v}) + n^2(\Gamma - \Lambda) + \frac{\rho(\mathbf{v} - \mathbf{v}_*)_\perp^2}{t_w}, \tag{11}$$

where $\rho$ is the density, **v** the velocity, $P$ the pressure, $\varepsilon$ the specific internal gas energy, $n$ the concentration, $\Gamma$ and $\Lambda$ the radiation heating and cooling functions. The term $2(\mathbf{v} \times \boldsymbol{\Omega})$ in the motion Equation (8) describes the Coriolis force.

The coefficient of magnetic turbulent viscosity $\eta_w$ is determined from the following expression:

$$\eta_w = \alpha_w \frac{l_w B_*}{\sqrt{4\pi\rho}} \tag{12}$$

where $\alpha_w$ is a dimensionless coefficient, which characterizes the wave turbulence efficiency, and $l_w = B_*/|\nabla B_*|$ is a characteristic spacial scale of wave pulsation. In our calculations, the value of $\alpha_w$ is equal to 1/3, which corresponds to isotropic turbulence [32].

The relaxation time scale $t_w$ for the transverse velocity component is as follows:

$$t_w = \frac{4\pi\rho\eta_w}{B_*^2}.$$  (13)

Note that in the case of synchronous polars, the velocity of magnetic field lines should be taken as $\mathbf{v}_* = 0$.

Density, internal energy and pressure are related by the equation of state of an ideal gas:

$$P = (\gamma - 1)\rho\varepsilon$$  (14)

where $\gamma = 5/3$—the adiabatic index.

The energy Equation (11) takes into account the effects of radiation heating and cooling [33–36], as well as heating due to current dissipation. In our numerical model, a linear approximation for the functions of radiation heating $\Gamma$ and cooling $\Lambda$ from the temperature $T$ in vicinity of its equilibrium value $T_*$ corresponding to the effective temperature of the hot component $T_a$ is used:

$$\Gamma = \Gamma_* + \Gamma'_*(T - T_*), \quad \Lambda = \Lambda_* + \Lambda'_*(T - T_*),$$  (15)

where $\Gamma_* = \Lambda_*$, $\Gamma'_*$ and $\Lambda'_*$ are constants, which values are set up individually for a specific polar. When the difference $\Gamma - \Lambda$ is calculated, the values $\Gamma_*$ and $\Lambda_*$ are being reduced.

The following initial and boundary conditions are used in the numerical model. In the envelope of the donor star, the normal velocity component with respect to its surface $v_n$ was set equal to the local speed of sound $c_s$, corresponding to the effective temperature of the donor $T_d$. The gas density in the donor envelope $\rho(L_1)$ is determined from the expression for the mass transfer rate through the internal Lagrange point $L_1$:

$$\dot{M} = \rho(L_1)v_n S_s$$  (16)

where $S_s$ is the cross-sectional area of the jet from the donor, and it is calculated by expression [4,13]:

$$S_s = \frac{\pi c_s^2}{4\Omega^2} g_y(q) g_z(q),$$  (17)

where $g_y(q)$ and $g_z(q)$ are dimensionless parameters depending on the mass ratio $q = M_d/M_a$ of the components of the binary system, specifying the large and small semi-axes, respectively, of the elliptical cross section of the jet. The accretor is defined by a sphere of radius $R_a$ on the boundary of which the free flow conditions are set. Constant conditions are applied at the external boundaries of the computational domain: density $\rho_b = 10^{-8}\rho(L_1)$, temperature $T_b = T_*$, magnetic field $\mathbf{b}_b = 0$. For the velocity $\mathbf{v}_b$, the free flow conditions are set: when the velocity is directed outward, the symmetric boundary conditions $\partial \mathbf{v}_b/\partial \mathbf{n} = 0$ are used, and when the velocity is directed inward, the conditions $\mathbf{v}_b = 0$ are used. The initial conditions in the computational domain are: density $\rho_0 = 10^{-8}\rho(L_1)$, temperature $T_0 = T_*$, velocity $\mathbf{v}_0 = 0$ and magnetic field $\mathbf{b}_0 = 0$.

For the calculations, a three-dimensional numerical code is used based on the Roe–Osher–Einfeldt difference scheme for magnetic hydrodynamics, which refers to Godunov-type schemes of an increased accuracy order. This difference scheme is described in detail in our paper [37]. The problem is solved in the computational domain $-2.0 \leq x/A \leq 1.0$, $-1.5 \leq y/A \leq 1.5$ and $-0.75 \leq z/A \leq 0.75$ with the number of cells $256 \times 256 \times 128$ and an uneven grid step exponentially decreasing toward the center of the accretor. Here the symbol A indicates the distance between the donor and accretor centers (see Figure 1). Such a computational domain completely includes the Roche lobes of the accretor and donor.

At the second stage of our investigation, the maps of temperature distribution over the accretor surface are constructed based on the results of flow structure numerical modeling, obtained at the first stage. The model assumes that the surface temperature of a white dwarf is the sum of two values: the effective temperature at a quiescent state $T_a$ and the temperature, which is proportional to the energy of the accreting matter. In the quiescent state, the accretor radiation is considered blackbody, and its effective flux is determined by the Stefan–Boltzmann law. In the case of matter accretion, the total flux is defined as a sum of effective flux and radiation into which the kinetic and thermal energy of the incident matter partially passes. The energy flux density of accreting matter at a point $\mathbf{R}_a$ of the accretor surface is determined by the following expression:

$$q(\mathbf{R}_a) = -\rho \mathbf{v} \cdot \mathbf{n}_a \left( \varepsilon + \frac{\mathbf{v}^2}{2} + \frac{P}{\rho} \right),\tag{18}$$

where $\mathbf{n}_a$ is the vector of the normal to the surface of the white dwarf. The minus sign takes into account the negative value of the normal component of the velocity of the incident matter: $\mathbf{v} \cdot \mathbf{n}_a < 0$. In this case, the value of the energy flux density $q(\mathbf{R}_a)$ turns out to be positive. Thus, the local temperature $T(\mathbf{R}_a)$ at the given point on the surface must satisfy the following relation:

$$T(\mathbf{R}_a) = \left[ T_a^4 + \frac{\kappa\, q(\mathbf{R}_a)}{\sigma_{\mathrm{SB}}} \right]^{1/4},\tag{19}$$

where $\sigma_{\mathrm{SB}}$ is the Stefan–Boltzmann constant, and $\kappa$ is the coefficient of processing the energy of accreting matter into radiation; according to the virial theorem, its value is assumed to be 0.5.

## 3. Calculation Results

### 3.1. Synchronous Polar

Calculations were performed for the typical synchronous polar V808 Aur, using the stationary model with the following initial parameters [38–48] (see Table 1):

**Table 1.** Initial parameters for calculations of the synchronous polar V808 Aur.

| Parameter | Symbol | Value | Unit |
|---|---|---|---|
| Donor mass | $M_d$ | 0.18 | $M_\odot$ |
| Accretor mass | $M_a$ | 0.86 | $M_\odot$ |
| Donor temperature | $T_d$ | 3400 | K |
| Accretor temperature | $T_a$ | 14,000 | K |
| Accretor radius | $R_a$ | 0.01375 | $R_\odot$ |
| Major axis of the orbit | A | 0.8 | $R_\odot$ |
| Inclination of the orbital plane | i | 79 | ° |
| Orbital period | $P_{\mathrm{orb}}$ | 1.95 | hour |
| Magnetic field induction at magnetic pole region | $B_a$ | 38 | MGs |
| Magnetic axis latitude angle | $\theta$ | 33 | ° |
| Magnetic axis longitude angle | $\phi$ | 170 | ° |

Four variants (models) of calculations of the flow structure are performed, which differ in values of the mass transfer rate. This parameter is set to: Model 1—$\dot{M} = 10^{-7}\ M_\odot/\text{year}$, Model 2—$\dot{M} = 10^{-8}\ M_\odot/\text{year}$, Model 3—$\dot{M} = 10^{-9}\ M_\odot/\text{year}$ and Model

4—$\dot{M} = 10^{-10} \ M_\odot/\text{year}$. It is assumed that Model 1 and Model 2 correspond to the high, Model 3 to the middle, and Model 4 to the low states of the polar.

The results of the three-dimensional numerical calculations for the synchronous system are presented in Figures 2 and 3. The figures show the iso-surfaces of the density logarithm in units of the density value at the Lagrange point $L_1$. It should be taken into account that when constructing each panel, an individual density scale is used relative to a given value $\rho(L_1)$ since $\rho(L_1)$ depends on the mass transfer rate.

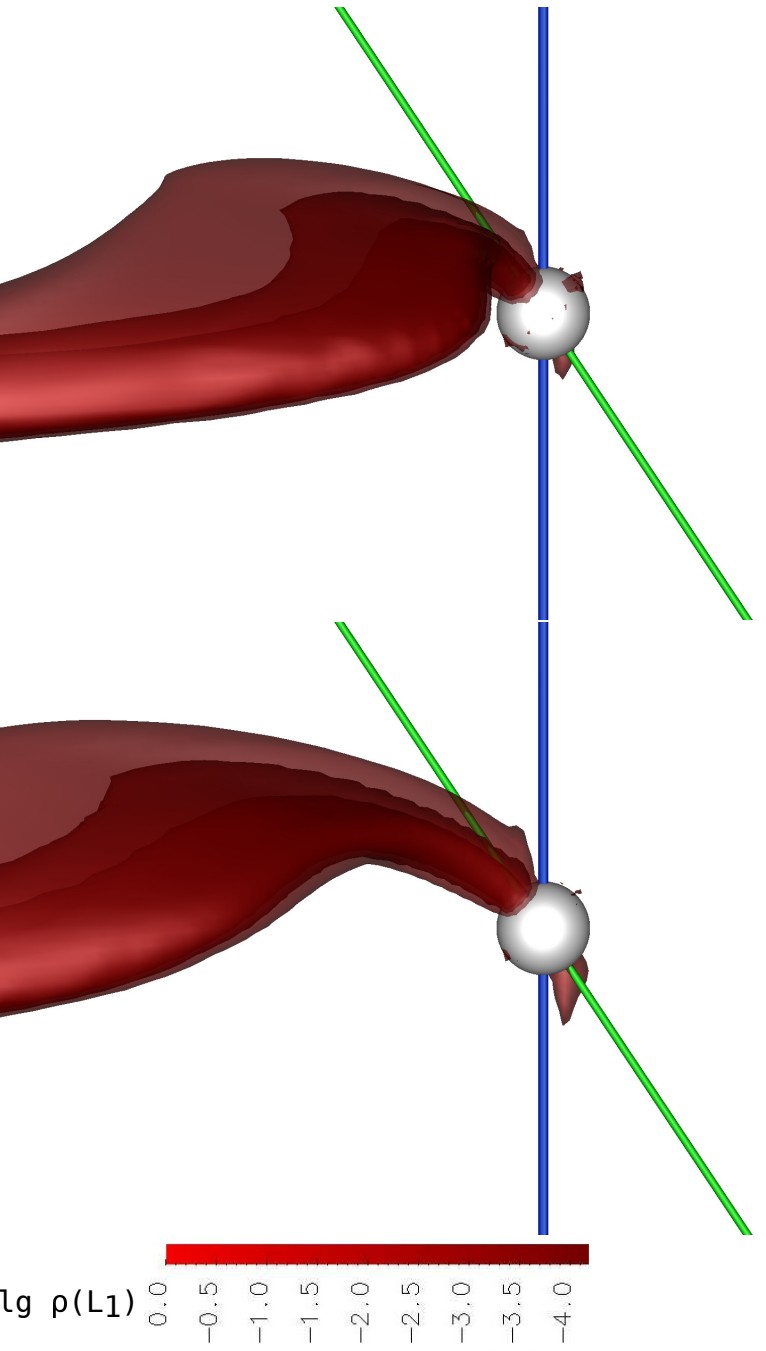

**Figure 2.** The result of three-dimensional numerical modeling of the flow structure in the synchronous polar for Model 1 (**upper** panel) and Model 2 (**lower** panel). The iso-surfaces of the decimal logarithm of the density in units of $\rho(L_1)$ are shown. The accretor is represented as a white sphere. The blue vertical line passing through the accretor coincides with its axis of rotation, and the inclined green line with its magnetic one. The binary system is shown at the point of view of an Earth observer at the orbital phase 0.8.

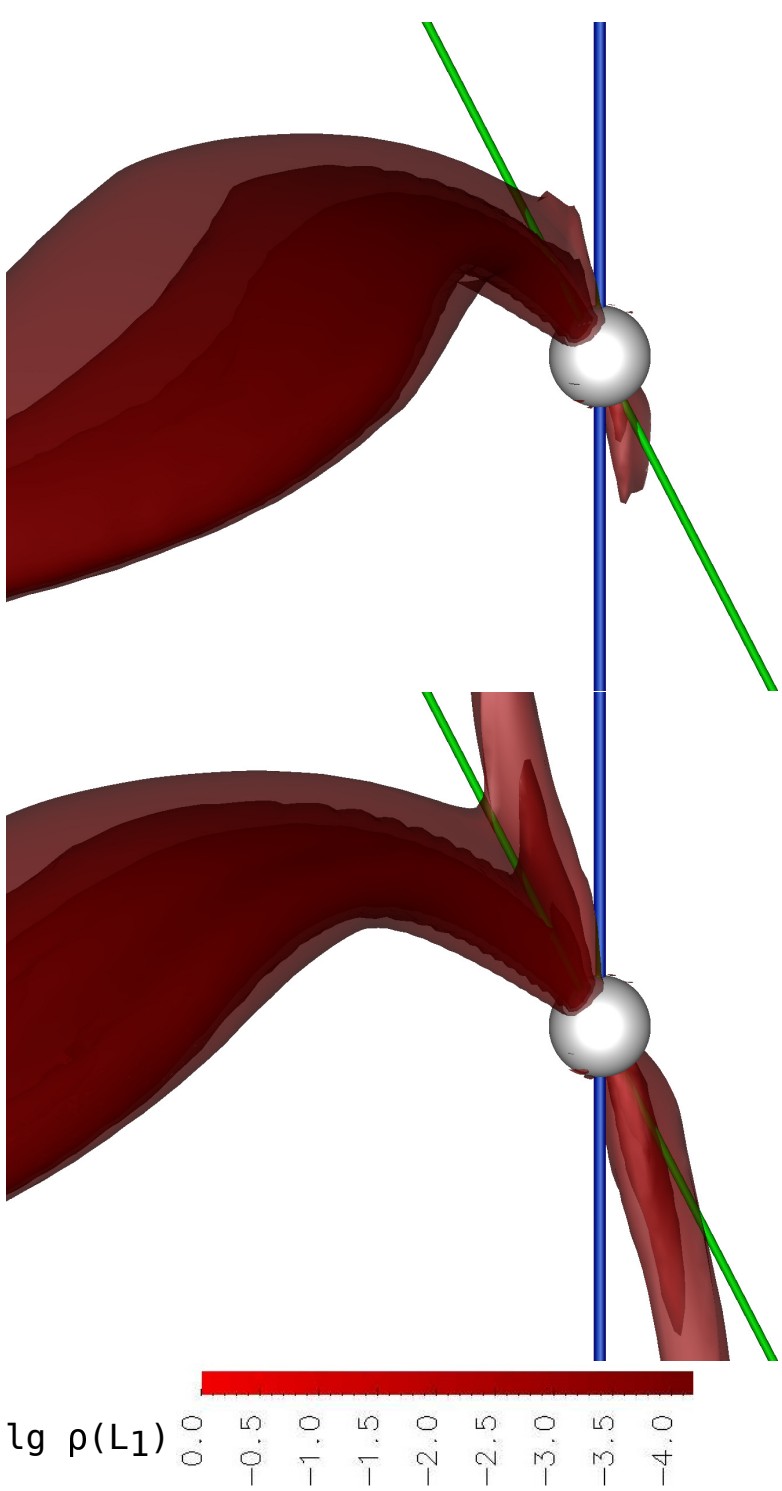

**Figure 3.** The same as Figure 2, but for Model 3 (**upper** panel) and Model 4 (**lower** panel). The binary system is shown from the point of view of an Earth observer at the orbital phase 0.9.

In Figure 2, numerical solutions for Model 1 and Model 2 are presented. The figure shows that the changes in the flow structure are insignificant for both models. The density of the jet matter near the Lagrange point $L_1$ differs by an order of magnitude, but remains quite high: for Model 1, $\rho(L_1) = 1.2 \times 10^{-5}$ g/cm$^3$, and for Model 2, $\rho(L_1) = 1.2 \times 10^{-6}$ g/cm$^3$. This leads to the fact that almost to the accretor itself, the jet matter moves along the ballistic trajectory. At the same time, a noticeable separation of matter by density is observed in the flow structure. The densest layers deviate more strongly from the direction of the primary

component, including by the action of the Coriolis force, while the movement of layers with lower density is controlled mainly by the magnetic field of the accretor. It is obvious that for this flow configuration, the accretion of the densest layers should occur at some distance from the magnetic pole, which is observed in the presented solutions (see also Figure 4).

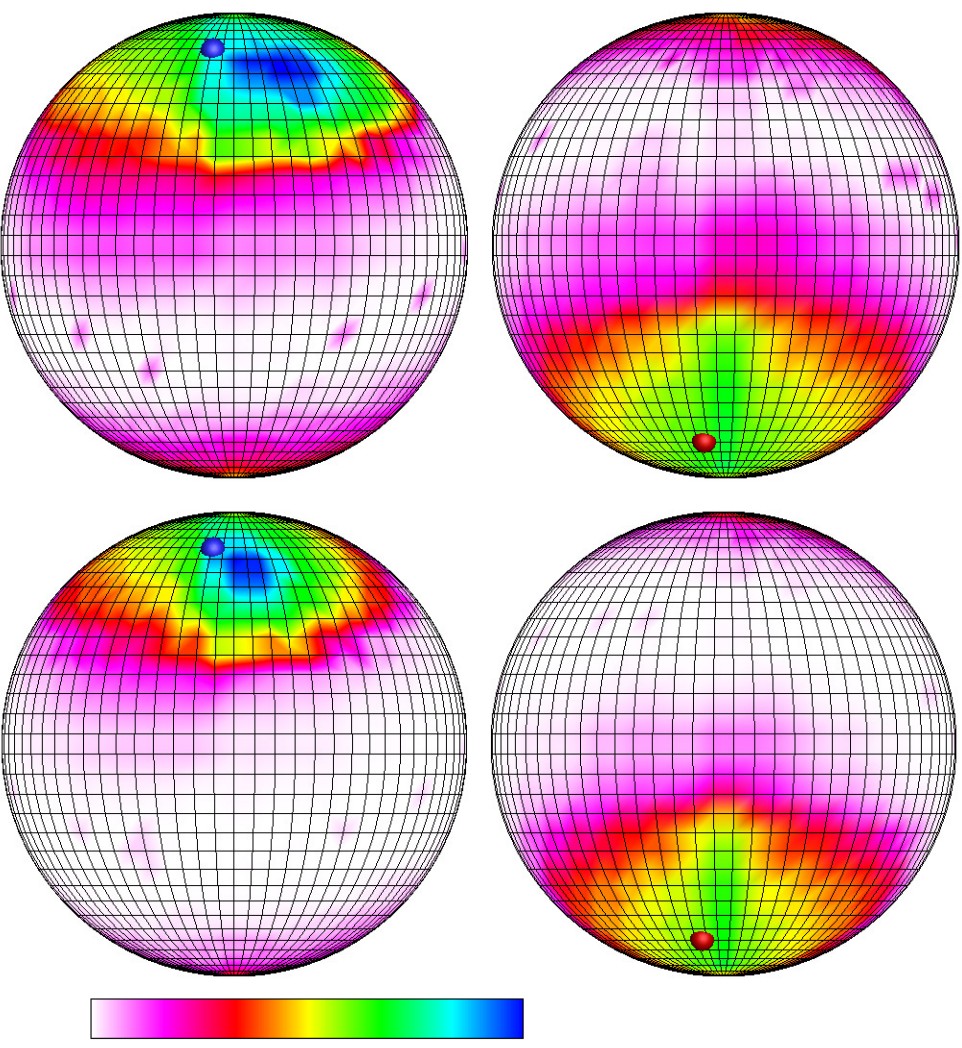

**Figure 4.** Temperature distribution over the accretor surface for Model 1 (**upper** panel) and Model 2 (**lower** panel). The **left** hemisphere corresponds to the side of the accretor facing the donor; the **right** hemisphere corresponds to the opposite side of the accretor. The symbols on the surface of the accretor indicate the positions of the north (blue) and south (red) magnetic poles, taking into account the orientation of the magnetic dipole relative to the donor.

A further decrease in the mass transfer rate and the density $\rho(L_1)$ changes the ratio of the ballistic and magnetic parts of the trajectory by an order of magnitude. In Model 3 (Figure 3, upper panel), for which $\rho(L_1) = 1.2 \times 10^{-7}$ g/cm$^3$, these parts become approximately equal. The magnetic field of the star much more strongly deflects the motion of the jet and directs it closer to the magnetic pole (see also Figure 5). The jet matter in its motion rises above the equatorial plane of the polar and moves along the magnetic force lines.

In the low state of the system (Figure 3, lower panel, Model 4), the density of the jet near the Lagrange point $\rho(L_1) = 1.2 \times 10^{-8}$ g/cm$^3$. In this model, we can see all the structures marked in Model 3. Here, the ballistic part of the jet is practically absent. The matter is almost immediately captured up by the magnetic field and is carried along

its lines to the magnetic pole of the accretor. It is also possible to observe an increase in the flow of matter to the both magnetic poles from a common envelope of the binary system.

Note that in the presented calculations, the accretion of jet matter occurs only at one pole. The fact is that in the accepted configuration, this magnetic pole is located opposite the donor, so the matter, having barely left the vicinity of inner Lagrange point $L_1$, rushes immediately to it. As follows from the analysis of the full picture of the flow, accretion to the second pole (at the level of a few percent of the total mass transfer rate) comes from the common envelope of the binary system. As we assume, the formation of a common envelope can occur as follows. In our numerical model, there is a boundary condition—the density at the border of computational domain, which is small, but not equal to zero. It is based on the fact that even in polars, there is some rarefied matter between components in the system. This matter could be considered a common envelope in the binary system. Its formation enables two mechanisms: due to part of the stream from the inner Lagrange point $L_1$, which does not include the main mass transfer flow, but dissipates in the system, and due to accreting of the matter flowing from the outer point $L_2$ under the action of dissipative processes.

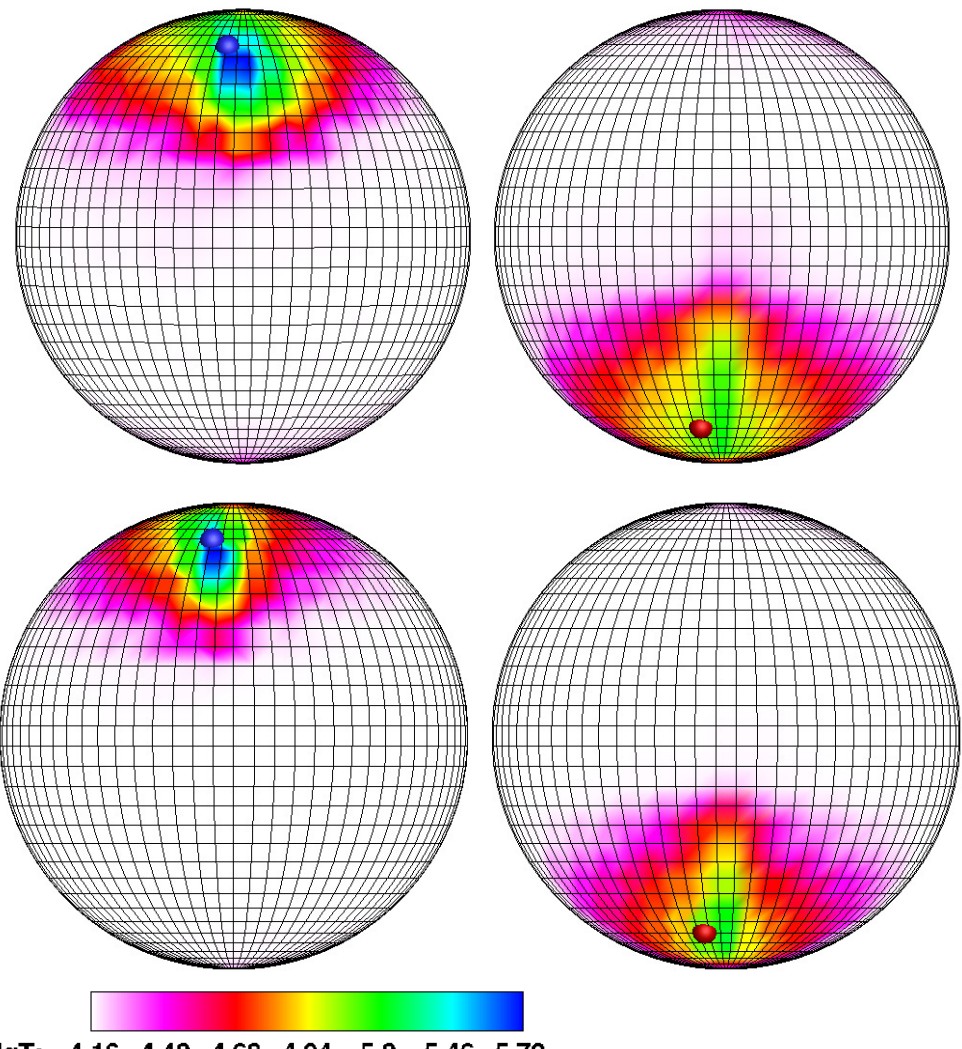

**Figure 5.** The same as in Figure 4, but for Model 3 (**upper** panel) and Model 4 (**lower** panel).

In Figures 4 and 5, the calculated temperature distributions over the accretor surface for the same models as in Figures 2 and 3 are presented. For clarity, a grid with a step in longitude and latitude of 5° is applied to the surface of the accretor. The left hemisphere

of each panel contains a white dwarf hemisphere with a north magnetic pole facing the donor. The opposite hemisphere of the white dwarf, containing the south magnetic pole, is represented on the right panel. Positions of the north and south magnetic poles are marked with blue and red symbols, respectively. On the logarithmic temperature scale shown at the bottom of figures, the white color corresponds to the effective temperature of white dwarf $T_a = 14,000$ K, and blue one corresponds to the temperature of the hot spot (of the order $10^6$ K).

The figures show that the energy release zones are concentrated near the north (main zone) and south (secondary zone) magnetic poles. Its intensity is different: the northern zone, corresponding to the accretion of matter from the jet, has a temperature about five times higher than the southern one, and this ratio is maintained when the mass transfer rate changes. It is also obvious that a decrease in the accretion rate leads to a narrowing of the areas of elevated temperature and a corresponding decrease in the area of the hot spot.

The position of hot spots is determined by the nature of accretion. Thus, the location of the southern hot spot does not depend on the accretion rate since it is formed by incident matter from the common envelope, not by the matter flow of the donor.

The location of the northern hot spot, on the contrary, significantly depends on the accretion rate. With an increase in the mass transfer rate $\dot{M}$, the jet density grows, and the flow, due to the action of inertia forces, is shifted to a greater extent in the direction opposite to the orbital rotation of the accretor. It leads, as discussed above, to a shift of the hot spot from the magnetic pole to the right in longitude and down in latitude. So, for Model 1 (Figure 4, upper panel), we can conclude that the hot spot is shifted in longitude relative to the north magnetic pole by about 30°. The latitude offset in this case is 5–7°. In Model 2 (Figure 4, lower panel), the offset in longitude decreases to 15°, while the value of the latitude of the hot spot practically does not change, but the area of the energy release zone is reduced by two times. In the middle state of the polar (Model 3, Figure 5, upper panel), the northern spot, while maintaining its geometric dimensions, is closely approaching the magnetic pole. At the same time, its shifts in longitude and latitude are 5°. In the low state (Model 4, Figure 5, lower panel), the northern region of energy release practically coincides with the northern magnetic pole; its area is decreased by four times compared to Model 1.

### 3.2. Asynchronous Polar

In addition to change in the mass transfer rate, another reason for the movement of hot spots is a variation in the configuration of the accretor magnetic field. A similar phenomenon can be observed in asynchronous polars. Due to alteration in the position of the magnetic dipole axis over time, and hence, the magnetic poles relative to the donor, in these objects, there is a sequential switching of field lines along which the jet matter accretes to the primary component. On time scale of the spin-orbital period in such a binary system, four accretion stages are distinguished. The first two of them are associated with the fallout of jet matter on one of the magnetic poles. Changing the position of the pole is a slow process (the characteristic lifetime of each configuration is about 40 orbital periods of the system; such a value is calculated by Formula (1) and the initial data), which allows us to consider the solution stationary at every moment of time and use the results of the calculations obtained for the synchronous polar to describe the drift of the hot spot (see Section 3.1). The motion of the hot spot caused by the pole shift and the corresponding change in the magnetic part of trajectory can be described analytically.

The other two stages of accretion are the processes of switching the jet between the magnetic poles [49,50]. In this case, there is a change in the ballistic part of the jet caused by variations in the accretion rate on both spots and in the magnetic configuration, which together lead to a complex picture of hot spots movement. In this section, we analyze the calculation results of the jet switching processes between magnetic poles in the asynchronous polar for various initial configurations of the magnetic field. It is assumed that the accretor has a dipole field configuration, but the dipole axis can be shifted relative to the center of the white dwarf.

The calculations are performed using the non-stationary [49] model for the typical asynchronous polar CD Ind with the following initial parameters [51–63] (see Table 2).

**Table 2.** Initial parameters for calculations of the asynchronous polar CD Ind.

| Parameter | Symbol | Value | Unit |
|---|---|---|---|
| Donor mass | $M_\mathrm{d}$ | 0.21 | $M_\odot$ |
| Accretor mass | $M_\mathrm{a}$ | 0.7 | $M_\odot$ |
| Donor temperature | $T_\mathrm{d}$ | 3200 | K |
| Accretor temperature | $T_\mathrm{a}$ | 12,000 | K |
| Accretor radius | $R_\mathrm{a}$ | 0.014 | $R_\odot$ |
| Major axis of the orbit | A | 0.735 | $R_\odot$ |
| Inclination of the orbital plane | i | 70 | ° |
| Orbital period | $P_\mathrm{orb}$ | 1.84 | hour |
| Accretor spin period | $P_\mathrm{spin}$ | 1.82 | hour |
| Beat period | $P_\mathrm{beat}$ | 174.5 | hour |
| Magnetic field induction on surface of a white dwarf (from observations) | $B_\mathrm{a}$ | 11 | MGs |
| Magnetic axis latitude angle | $\theta$ | 70 | ° |
| Magnetic axis longitude angle | $\phi$ | 90 | ° |

In our paper, we consider two models of the asynchronous polar CD Ind based on its observational data.

The first model corresponds to the really observed system and, according to observer's estimates, there is a configuration of a magnetic field with an offset dipole in it. The magnitude of this displacement is half of the radius of the accretor beneath the orbital plane. In this case, we understand that the real magnetic field could be more complex, but as it is obvious for observers, the real field can be described by some effective model of an offset dipole.

In the second model, when considering the same polar, we artificially move the axis of the magnetic dipole so that it passes through the center of the accretor. Thus, we obtain the configuration of the field with a non-offset dipole. This model is used as a comparison with the synchronous polar considered in Section 3.1 since by the same central position of the magnetic axis in these polars, there are different mechanisms that impact the movement of the hot spots. So, if in the synchronous polar, as we have seen, the drift of hot spots is determined by a change in the mass transfer rate, then in the asynchronous polar, this is caused by a change in the position of the magnetic axis in time during asynchronous rotation of the accretor.

The rate of mass transfer in both models is assumed to be unchanged and equal to $\dot{M} = 10^{-9}\ M_\odot/\text{year}$.

The results of three-dimensional numerical calculations for the variant with the non-offset dipole corresponding to switching of the flow from the south magnetic pole to the north one are presented in Figures 6 and 7. The graphical symbols used here are the same as in Figures 2 and 3. The figures illustrate four phases of the switching process: Phase 1 is the beginning of switching (Figure 6, upper panel), Phase 2 is the formation moment of the arc of matter in the accretor magnetosphere (Figure 6, lower panel), Phase 3 is the moment of maximum mutual convergence of hot spots (Figure 7, upper panel) and Phase 4

is the switching completion (Figure 7, lower panel). On all four panels, the binary system is represented for the position of the observer at the orbital phase 0.9.

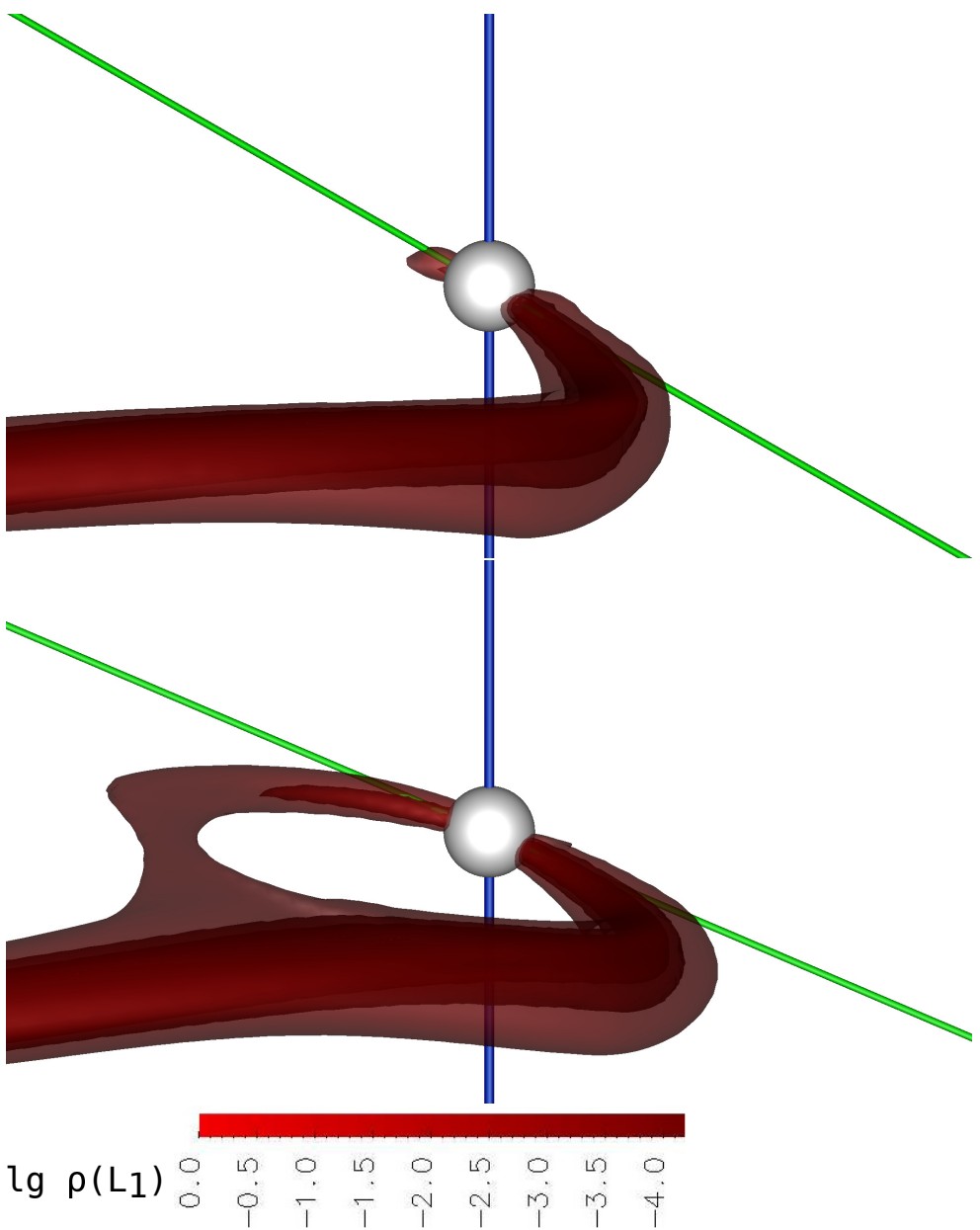

**Figure 6.** The result of three-dimensional numerical modeling of the structure of the flow matter in the asynchronous polar with the non-offset dipole for the moment of switching from the south magnetic pole to the north. The iso-surfaces of the decimal logarithm of the density are shown in units of $\rho(L_1)$. The surface of the accretor is represented as a white sphere. The binary system is at the orbital phase 0.9. The upper panel corresponds to Phase 1, and the lower one corresponds to Phase 2 of the switching process.

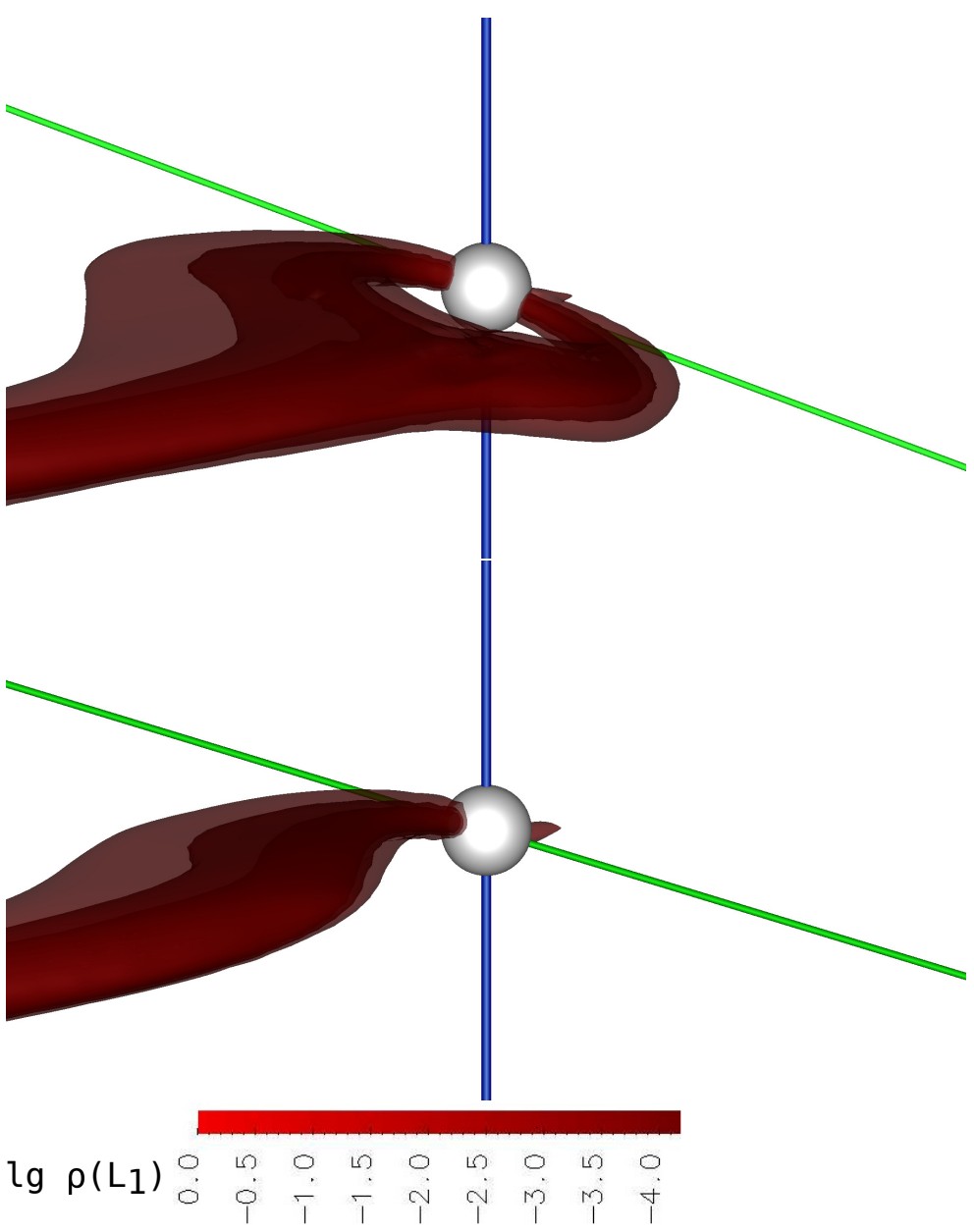

**Figure 7.** The same as in Figure 6, but for Phase 3 (**upper** panel)) and Phase 4 (**lower** panel).

The beginning of the switching (Figure 6, upper panel) coincides with the 1st orbital period from zero phase of the beat period and is characterized by the accretion of the jet just to the south magnetic pole. In the flow structure, it can be seen that the ballistic part of the trajectory for a given mass transfer rate has a longer length than the magnetic one. This flow configuration is similar to the jet geometry for the synchronous polar in a high state (see Figure 2); however, for the asynchronous polar in the middle state, this similarity is determined by the fact that the axis of the magnetic dipole is perpendicular to the line connecting the donor and accretor centers.

The stage of formation of an arc of matter in the magnetosphere (Figure 6, lower panel) corresponds to the 5th orbital period and is characterized by two features in the flow structure. First, an accretion of matter occurs at both magnetic poles, and the flow density at a given time to the north pole is an order of magnitude less than to the south. Secondly, the separation of the initially single jet into two streams leads to the formation of the arch of matter at the boundary of the magnetosphere of the white dwarf. Compared with a single-pole accretion (Figure 6, upper panel), when a split flow is formed, the jet

trajectory changes. The length of its ballistic part decreases to the upper boundary of the magnetosphere and then the matter continues to flow along the magnetic field lines: less dense layers accrete in vicinity of the north magnetic pole, and denser layers retain the accretion mode to the south pole.

With the orbital rotation of the polar, the size of the arc of matter in the magnetosphere undergoes changes. At the 9th orbital period, its outer radius still coincides with the boundary of the magnetosphere, and the inner one decreases, which leads to mutual convergence of the hot spots (Figure 7, upper panel). Obviously, in this case, we should expect the maximum deviation of spots from the magnetic poles. An increase in the volume of the arch indicates that there is an accumulation of matter in it. This, in turn, should be accompanied by an increase in the density of matter in the jet and a short-term decrease in the accretion rate to both poles.

Upon completion of the switching (Figure 7, lower panel), which occurs at the 13th orbital period, the ballistic part of the jet is significantly reduced, and the flow configuration is an analogy of the picture for the synchronous polar at low state (see Figure 3, lower panel). At this point of time, the north magnetic pole of the accretor is opposite the donor, and on the magnetic part of the trajectory, the jet matter rises slightly above the orbital plane of the polar, following the magnetic lines to the pole. As it can be seen from the figure, there is practically no deviation of the accretion region from the magnetic pole.

In Figures 8 and 9, the iso-surfaces of the density logarithm for the calculation variant with the non-offset dipole corresponding to the switching of the flow from the north magnetic pole to the south are shown. For clarity, the flow structure in the binary system is shown from the point of view of an Earth observer at the orbital phase 0.4.

As can be seen from the figure, the same elements are observed in the flow structure in the previous switching. As the corresponding phases of the switching processes, that are shown in Figures 6–9, coincide in time, we should expect similarity in the flow geometry. Indeed, the passage of the dipole axis through the center of the accretor should lead to equal induction values of both magnetic poles, and hence, to symmetry in the flow structure. The differences when switching the flow from the north magnetic pole to the south consist only in changing the orientation of the dipole axis and the corresponding change in the plane of matter motion in the magnetosphere. However, the physical parameters of the accretion zones and the flow itself near the accretor surface may differ, due to the orbital rotation of the polar. In particular, the Coriolis force should lead to differentiation of the values of the individual accretion rates to the northern and southern hot spots since in this case, one of the magnetic poles turns out to be somewhat closer to the Lagrange point $L_1$ than the other in terms of the length of the jet trajectory.

Below in Table 3, the calculated values of the individual accretion rates for switching flow from the south magnetic pole to the north one are given, and in Table 4, for switching flow from the north pole to the south one. For the notation of values in the tables, as before, $\rho(L_1)$ is the density of matter at the Lagrange point $L_1$; $\dot{M}_N$ is the individual accretion rate for the northern hot spot; $\dot{M}_S$ is individual accretion rate for the southern hot spot; $\dot{M}_{total} = \dot{M}_N + \dot{M}_S$ is the total accretion rate to both spots at a given value of the mass transfer rate $\dot{M}$. For Phase 1 and Phase 4, the inverse value is indicated in parentheses in the last column in order to compare both switches.

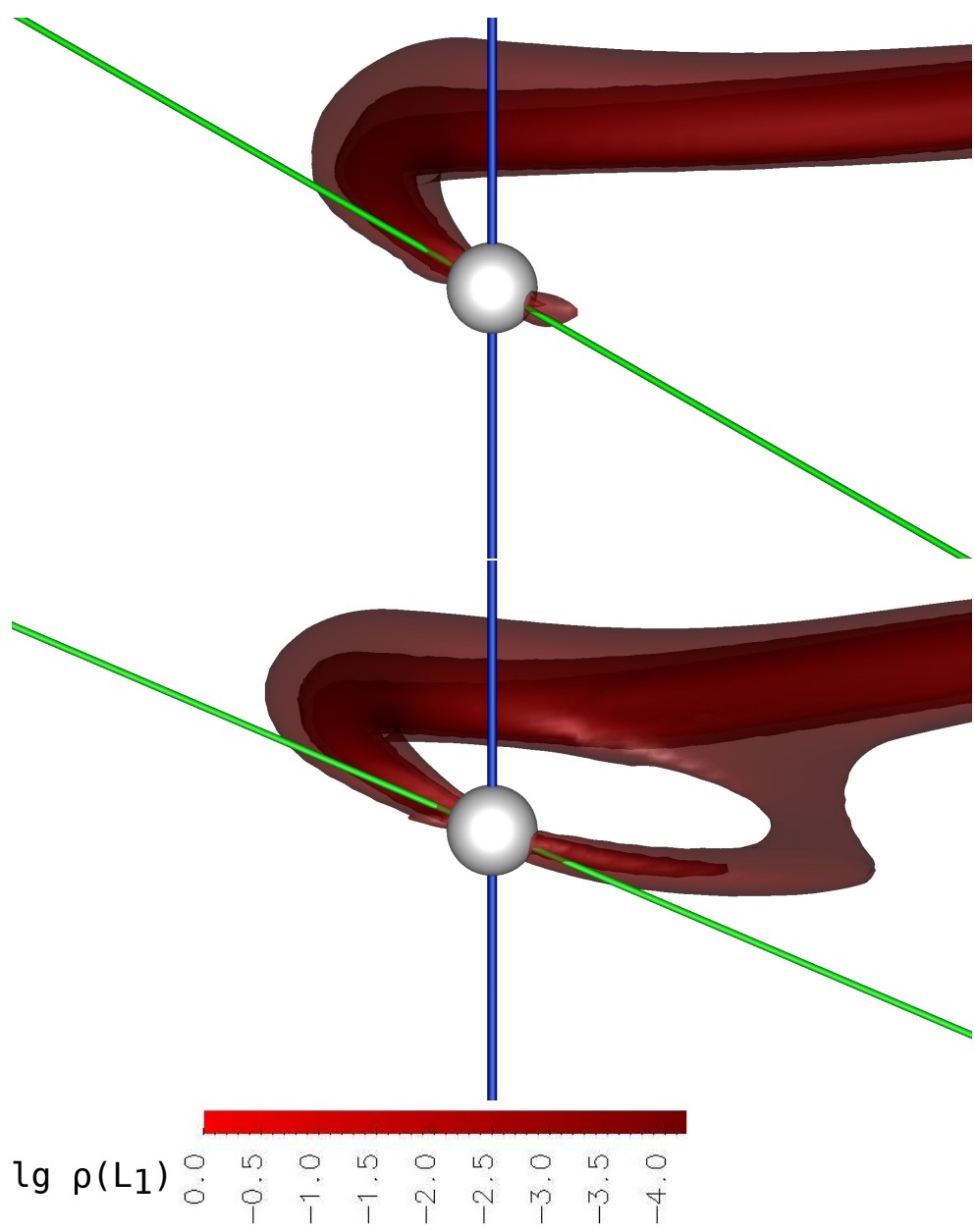

**Figure 8.** The result of three-dimensional numerical modeling of the structure of the flow of matter in an asynchronous polar with the non-offset dipole for the moment of switching from the north magnetic pole to the south one. The iso-surfaces of the decimal logarithm of the density are shown in units of $\rho(L_1)$. The surface of the accretor is represented as a white sphere. The binary system is at the orbital phase 0.4. The upper panel coincides with Phase 1, and the lower panel with Phase 2 of the switching process.

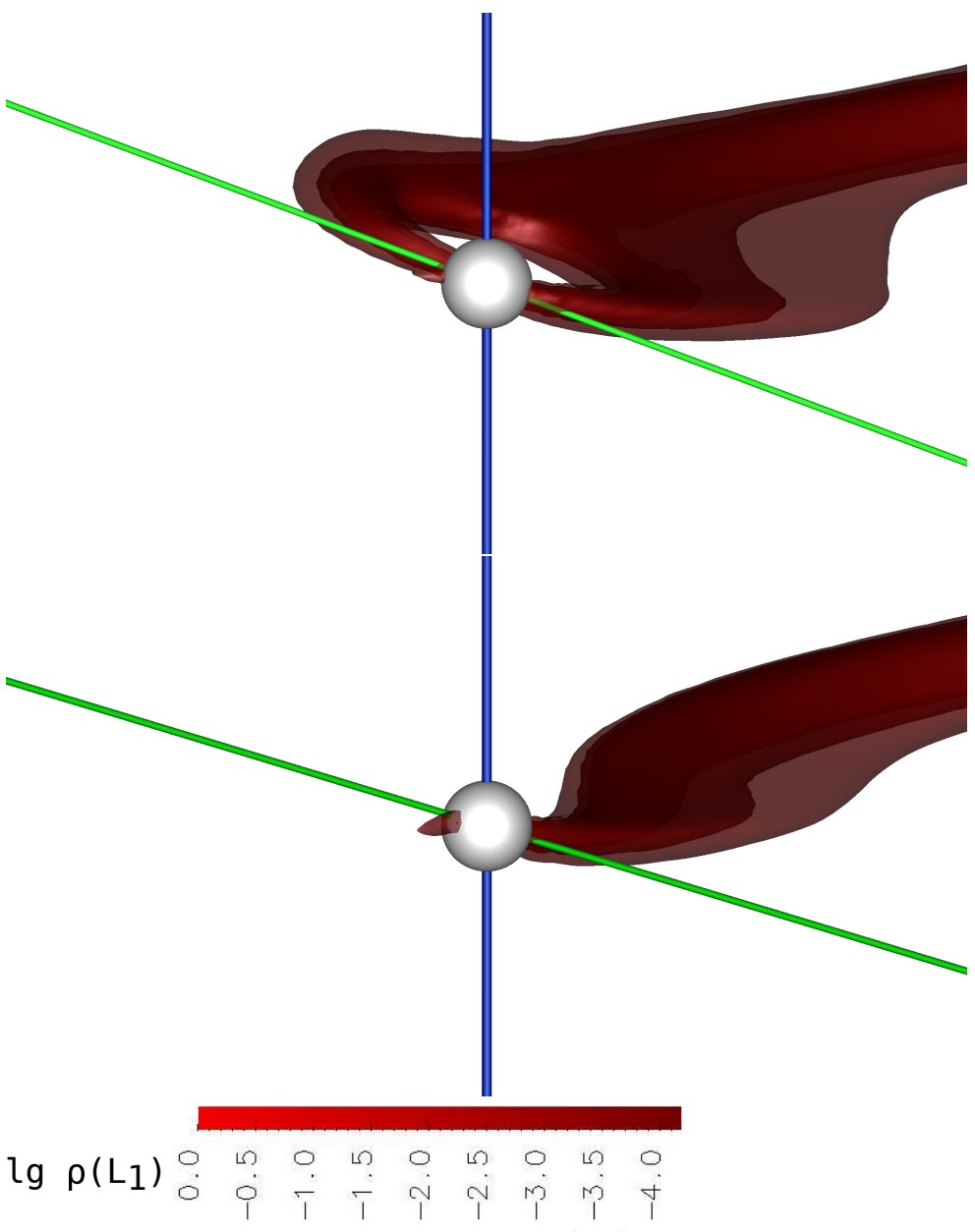

**Figure 9.** The same as in Figure 8, but for Phase 3 (**upper** panel)) and Phase 4 (**lower** panel).

**Table 3.** Parameters of accretion of matter on a white dwarf in the asynchronous polar with non-offset dipole for different phases of switching the flow from the south pole to the north one.

| Phase | $\rho(L_1)$, g/cm$^3$ | $\dot{M}_N$, $M_\odot$/year | $\dot{M}_S$, $M_\odot$/year | $\dot{M}_{total}$, $M_\odot$/year | $\dot{M}_N/\dot{M}_S$ $(\dot{M}_S/\dot{M}_N)$ |
|---|---|---|---|---|---|
| Phase 1 | $1.92 \times 10^{-7}$ | $2.32 \times 10^{-11}$ | $4.70 \times 10^{-9}$ | $4.72 \times 10^{-9}$ | $0.005\,(202.58)$ |
| Phase 2 | $1.92 \times 10^{-7}$ | $3.95 \times 10^{-10}$ | $3.97 \times 10^{-9}$ | $4.36 \times 10^{-9}$ | $0.099$ |
| Phase 3 | $1.92 \times 10^{-7}$ | $2.39 \times 10^{-9}$ | $1.44 \times 10^{-9}$ | $3.83 \times 10^{-9}$ | $1.659$ |
| Phase 4 | $1.92 \times 10^{-7}$ | $3.84 \times 10^{-9}$ | $9.95 \times 10^{-11}$ | $3.93 \times 10^{-9}$ | $38.59\,(0.025)$ |

**Table 4.** Parameters of accretion of matter on the white dwarf in the asynchronous polar with non-offset dipole for various phases of switching the flow from the north pole to the south one.

| Phase | $\rho(L_1)$, g/cm$^3$ | $\dot{M}_N$, $M_\odot$/year | $\dot{M}_S$, $M_\odot$/year | $\dot{M}_{total}$, $M_\odot$/year | $\dot{M}_N/\dot{M}_S$ ($\dot{M}_S/\dot{M}_N$) |
|---|---|---|---|---|---|
| Phase 1 | $1.92 \times 10^{-7}$ | $3.97 \times 10^{-9}$ | $1.45 \times 10^{-10}$ | $4.12 \times 10^{-9}$ | $27.37(0.036)$ |
| Phase 2 | $1.92 \times 10^{-7}$ | $3.12 \times 10^{-9}$ | $7.54 \times 10^{-10}$ | $3.88 \times 10^{-9}$ | $0.241$ |
| Phase 3 | $1.92 \times 10^{-7}$ | $1.2 \times 10^{-9}$ | $3.56 \times 10^{-9}$ | $4.76 \times 10^{-9}$ | $2.966$ |
| Phase 4 | $1.92 \times 10^{-7}$ | $3.52 \times 10^{-11}$ | $4.78 \times 10^{-9}$ | $4.82 \times 10^{-9}$ | $0.007(135.79)$ |

An analysis of the data presented in the tables shows that both switching processes differ primarily in the ratio of individual accretion rates (the last column). The most significant difference is observed at the initial stage of switching (Phase 1): for the switching from south pole to the north one, this ratio is seven times lower than for reverse switching. This value is determined by the smaller, by an order of magnitude, accretion of matter from the common envelope in the first case, while the accretion of matter from the jet to the active pole is approximately comparable in both cases. When an arch is formed in the magnetosphere and its volume increases (Phase 2 and Phase 3), the ratio of the accretion rates to both spots has insignificant differences. At the end of the switching (Phase 4), the ratio of the accretion rates differs by 3.5 times, which is again associated with a drop in the accretion of matter from the common envelope at the second switching.

In Figures 10–13, the results of numerical simulation of the system with an offset dipole are shown.

It can be seen from the figures that the displacement of the dipole practically does not affect the flow geometry. The position of the ballistic part of the jet trajectory remains unchanged compared to the configuration of the non-offset dipole. On the magnetic part of the trajectory, the plane of flow motion has a greater inclination in accordance with the magnitude of the dipole axis displacement. In addition, in the accretion structure, it can be observed the accumulation of matter on the surface of the white dwarf in the plane of the magnetic equator. This element is determined by action of a magnetic trap effect. Due to the fact that the magnetic field strength on the lower side of the accretor is higher than in the upper one, a part of the accreting matter is pushed into the region of the magnetic equator due to the energy loss. Making damped oscillations between two magnetic poles, this matter stops in the specified area. On the upper side of the accretor, the magnetic field of the same intensity is "hidden" inside the star, so there is no accumulation of matter at the equator here.

In order to compare the accretion parameters with the variant of the non-offset dipole, we also construct the corresponding tables for the case of the offset dipole. In Table 5, the characteristics for the flow switching from the south magnetic pole to the north one are shown, and in Table 6, those for the switching from the north pole to the south one are presented.

**Table 5.** Parameters of matter accretion to the white dwarf in the asynchronous polar with the offset dipole for various phases of switching the flow from the south pole to the north one.

| Phase | $\rho(L_1)$, g/cm$^3$ | $\dot{M}_N$, $M_\odot$/year | $\dot{M}_S$, $M_\odot$/year | $\dot{M}_{total}$, $M_\odot$/year | $\dot{M}_N/\dot{M}_S$ ($\dot{M}_S/\dot{M}_N$) |
|---|---|---|---|---|---|
| Phase 1 | $1.92 \times 10^{-7}$ | $7.49 \times 10^{-10}$ | $4.43 \times 10^{-9}$ | $4.84 \times 10^{-9}$ | $0.166(6.024)$ |
| Phase 2 | $1.92 \times 10^{-7}$ | $7.25 \times 10^{-10}$ | $4.14 \times 10^{-9}$ | $4.86 \times 10^{-9}$ | $1.751$ |
| Phase 3 | $1.92 \times 10^{-7}$ | $4.20 \times 10^{-9}$ | $1.06 \times 10^{-9}$ | $5.27 \times 10^{-9}$ | $3.962$ |
| Phase 4 | $1.92 \times 10^{-7}$ | $1.23 \times 10^{-8}$ | $2.90 \times 10^{-10}$ | $1.26 \times 10^{-8}$ | $424.13(0.023)$ |

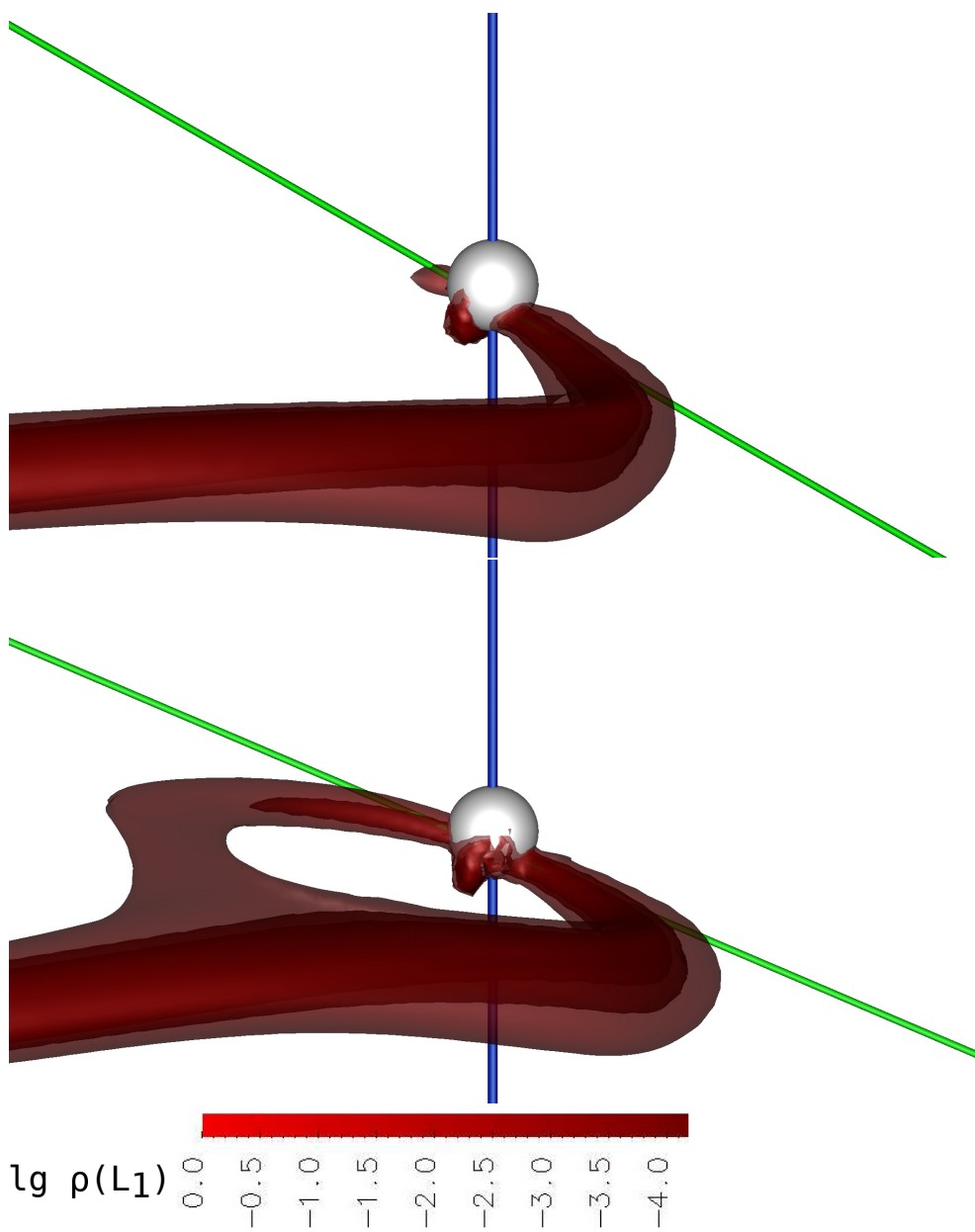

**Figure 10.** The three-dimensional numerical modeling results of the matter flow structure in the asynchronous polar with an offset dipole for the moment of switching from the south magnetic pole to the north one. The iso-surfaces of the decimal logarithm of the density are shown in units of $\rho(L_1)$. The surface of the accretor is presented as a white sphere. The binary system is at the orbital phase 0.9. The upper panel coincides with Phase 1, and the lower panel with Phase 2 of the switching process.

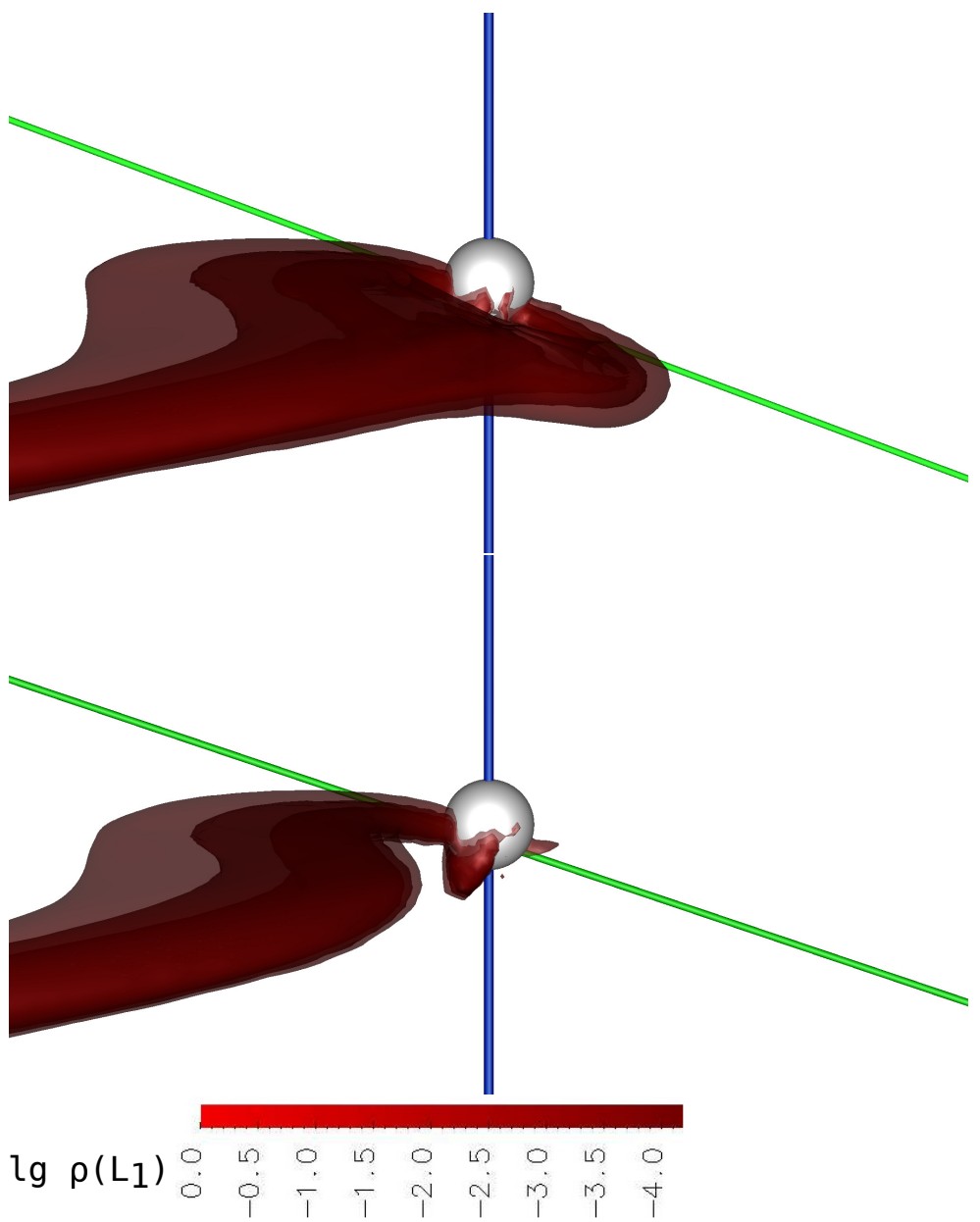

**Figure 11.** The same as in Figure 10, but for Phase 3 (**upper** panel)) and Phase 4 (**lower** panel).

**Table 6.** Parameters of matter accretion to the white dwarf in the asynchronous polar with offset dipole for various phases of switching the flow from the north pole to the south one.

| Phase | $\rho(L_1)$, g/cm$^3$ | $\dot{M}_N$, $M_\odot$/year | $\dot{M}_S$, $M_\odot$/year | $\dot{M}_{total}$, $M_\odot$/year | $\dot{M}_N/\dot{M}_S$ ($\dot{M}_S/\dot{M}_N$) |
|---|---|---|---|---|---|
| Phase 1 | $1.92 \times 10^{-7}$ | $6.06 \times 10^{-9}$ | $4.77 \times 10^{-10}$ | $6.53 \times 10^{-9}$ | $12.70(0.078)$ |
| Phase 2 | $1.92 \times 10^{-7}$ | $5.30 \times 10^{-9}$ | $8.84 \times 10^{-10}$ | $6.19 \times 10^{-9}$ | $5.995$ |
| Phase 3 | $1.92 \times 10^{-7}$ | $2.70 \times 10^{-9}$ | $3.25 \times 10^{-9}$ | $5.95 \times 10^{-9}$ | $0.830$ |
| Phase 4 | $1.92 \times 10^{-7}$ | $9.99 \times 10^{-10}$ | $4.64 \times 10^{-9}$ | $5.64 \times 10^{-9}$ | $0.204(4.901)$ |

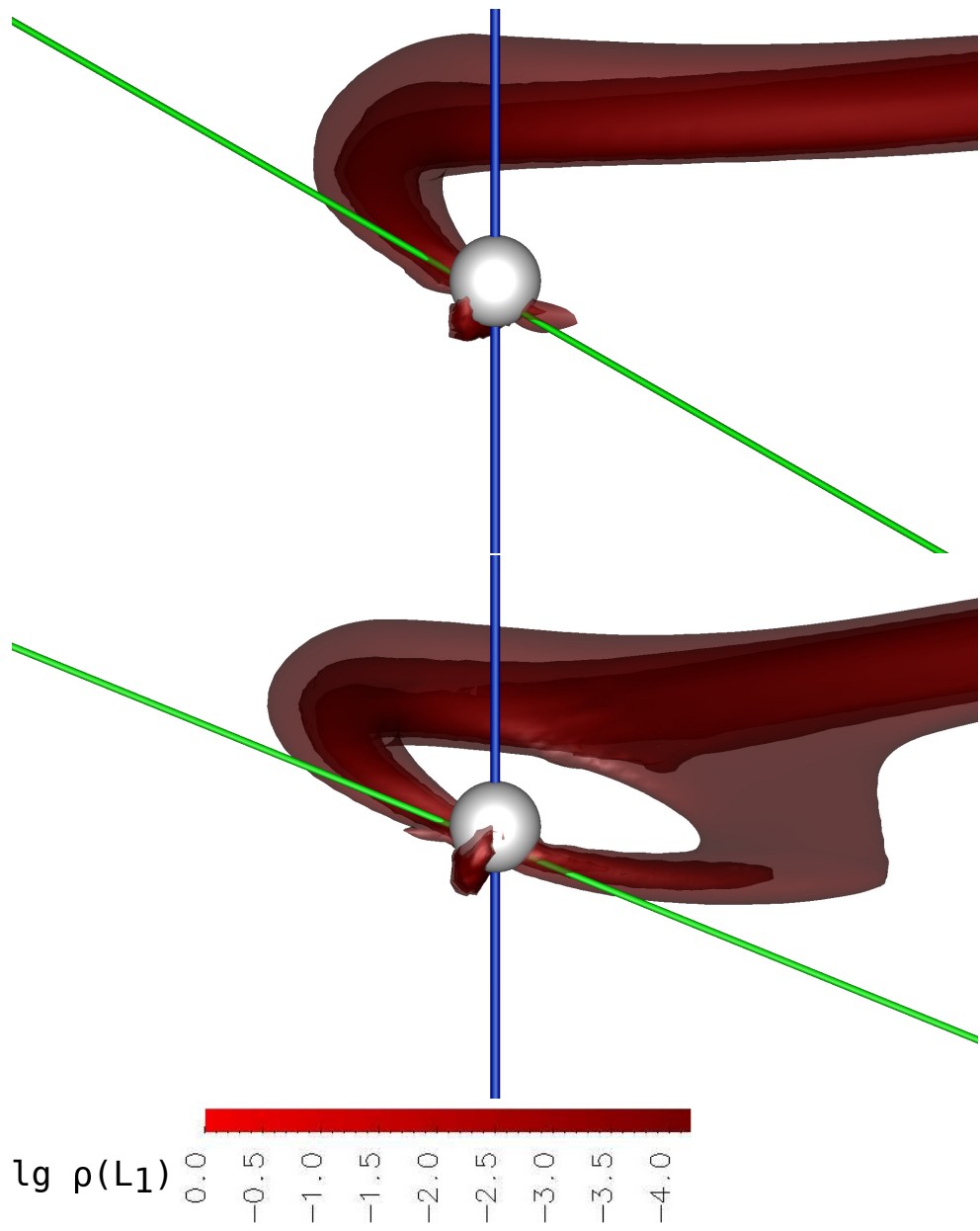

**Figure 12.** The three-dimensional numerical modeling results of the matter flow structure in the asynchronous polar with an offset dipole for the moment of switching from the north magnetic pole to the south one. The iso-surfaces of the decimal logarithm of the density are presented in units of $\rho(L_1)$. The surface of the accretor is shown as a white sphere. The binary system is at the orbital phase 0.4. The upper panel coincides with Phase 1, and the lower panel with Phase 2 of the switching process.

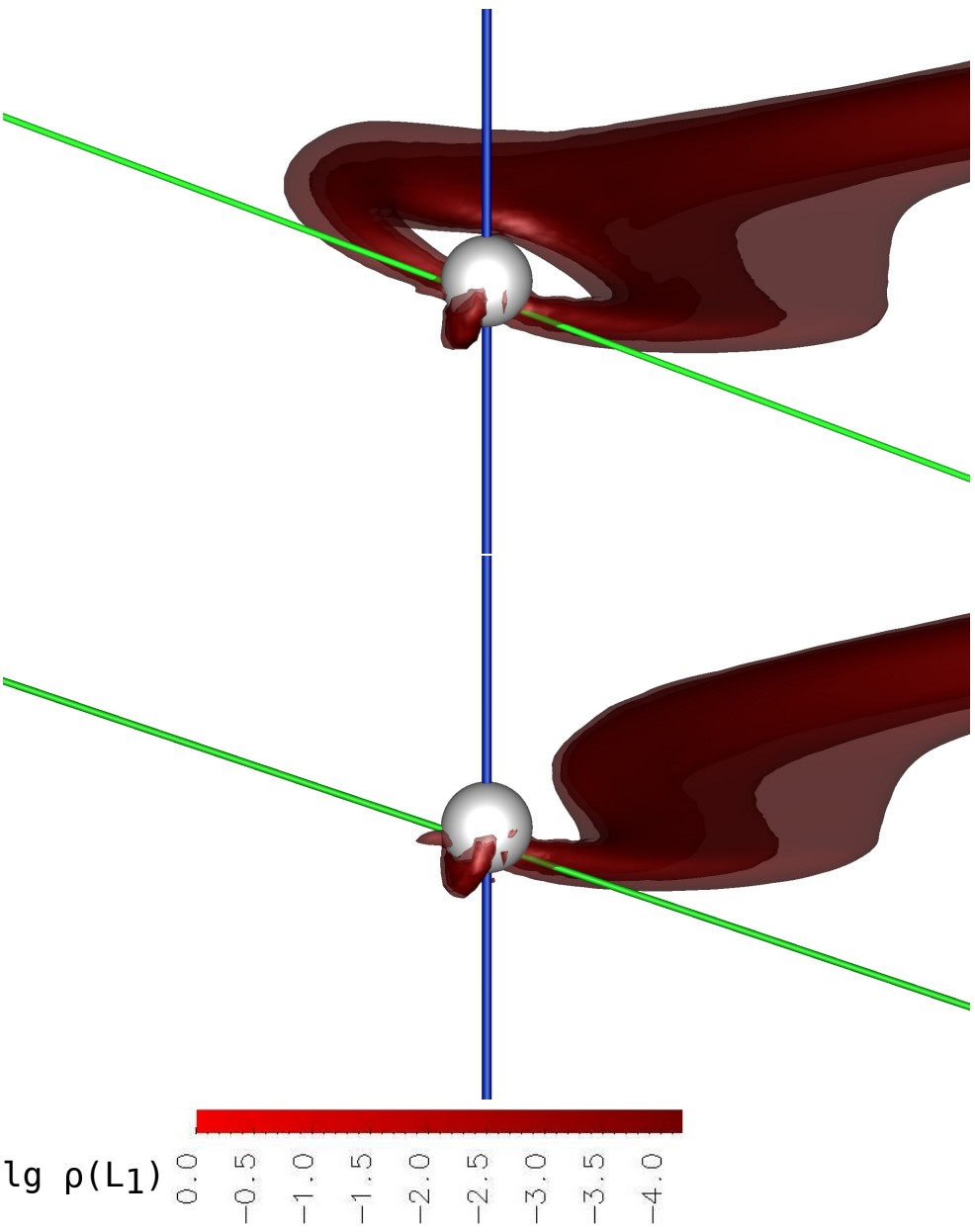

$lg \; \rho(L_1)$

**Figure 13.** The same as in Figure 12, but for Phase 3 (**upper** panel) and Phase 4 (**lower** panel).

It can be seen from the tables that the dipole displacement leads to different behavior of the total accretion rate for both switching processes. When the jet closes to the north magnetic pole in the first process, a sharp increase in the accretion rate is observed in Phase 4. This indicates the effect of matter accumulation in the jet during the formation of an arch in the magnetosphere of the white dwarf and its subsequent discharge to the magnetic pole. In addition, due to the shift of the dipole axis, the north pole is closer to the Lagrange point $L_1$ than the south one, which also contributes to a local increase in the accretion rate. When switching back from the north pole to the south one, the accretion rate experiences a slight drop, and the process of changing the magnetic poles proceeds smoothly. In this case, despite the fact that the field strength in the vicinity of the south pole is higher than the north one, the south pole is removed from the inner Lagrange point at a greater distance and is weaker from the point of view of accretion. If we compare the tempo values for the south pole in the first switch in Phase 1 and in the second switch in Phase 4, they turn out to be approximately equal. For the north pole in Phase 1, the accretion rate is initially slightly higher than for the south one, and after the first switching in Phase 4,

when the matter is discharged to this pole, the high velocity value is maintained for several orbital periods, while the pole is in the vicinity of the inner Lagrange point. It is worth noting that the accretion rate of matter from the polar common envelope does not undergo significant changes and is in the range (4–8) $\times 10^{-10}$ $M_\odot$/year for both poles.

In Figures 14–17, the results of calculating temperature maps of the accretor surface for the case of the non-offset magnetic dipole are shown.

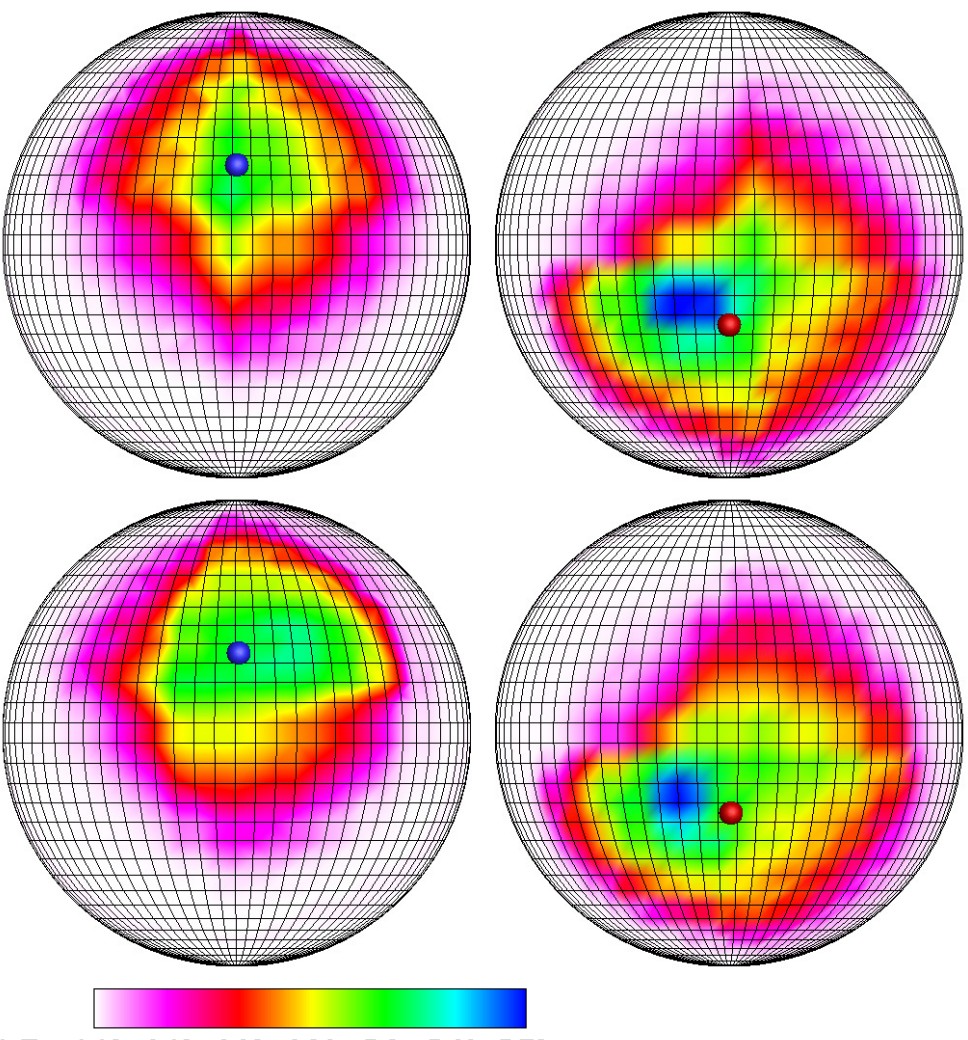

**Figure 14.** Temperature distribution over the accretor surface for the asynchronous polar with a non-offset dipole when switching the flow from the south magnetic pole to the north one. The temperature maps for Phase 1 (**upper** panel) and for Phase 2 (**lower** panel) are shown.

As it was noted above, this variant of the binary system is most interesting for comparison with the synchronous polar. The figures show that the deviation values of hot spots from the magnetic poles, determined by the processes of switching the flow between the poles and by the changing in the mass transfer rate, are comparable. However, the position of the spots in the asynchronous polar has a number of features. At the switching from the south pole to the north one in Phase 1 (Figure 14, upper panel), the hot spot is adjacent to the pole, which is observed in the flow pattern (Figure 6, upper panel) but it is elongated in longitude and has a maximum area. For the synchronous polar in the middle state (Figure 5, upper panel), i.e., at the same mass transfer rate, a similar spot is half as small. This indicates that with a fixed mutual arrangement of the donor and accretor in the synchronous system, the accretion zone tends to concentrate in a small area, whereas with asynchronous rotation and the placement of the magnetic pole close to the equator of the

primary component, the hot spot is distributed along the longitude in accordance with the density of the flow in the jet.

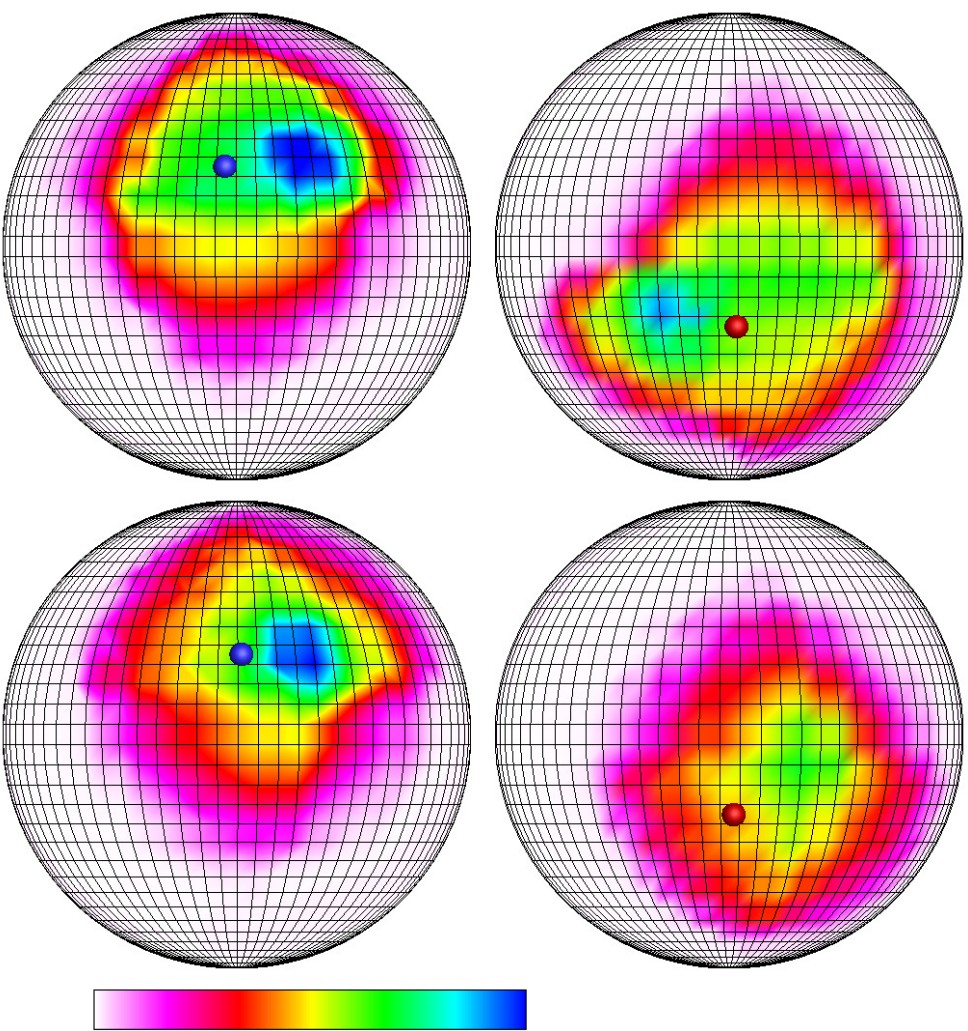

**Figure 15.** The same as in Figure 14, but for Phase 3 (**upper** panel) and Phase 4 (**lower** panel).

At the formation moment of the arc of matter (Figure 14, lower panel), the deviation of both spots from the magnetic poles does not exceed $15°$. At the same time, the area of the southern spot decreases by two times, and the temperature of the northern spot increases slightly and at this stage is still comparable to the surrounding area. The last circumstance is caused by the fact that although the density of the northern stream is approaching the southern one, the speed of this stream is relatively low at this stage.

The switching Phase 3 (Figure 15, upper panel) illustrates the moment of maximum convergence of hot spots. The deviation angle of both spots from the magnetic poles is increased and is approximately $20°$. It is worth noting that theoretically, in this case, the flows of matter to both poles should be approximately equal, and the spots should have a comparable temperature. However, as follows from Table 5, third line, this equality is not fulfilled: the accretion rates differ by 1.7 times at this phase, and the corresponding temperature map shows that the southern hot spot is noticeably weaker than the northern one. From Figure 7, upper panel, we can conclude that this is due to the orbital rotation of the polar. As a result of the approach of the north magnetic pole to the Lagrange point $L_1$ and the removal of the south pole from it, the ballistic part of the jet trajectory is somewhat reduced, and the most matter accretes mainly to the northern spot.

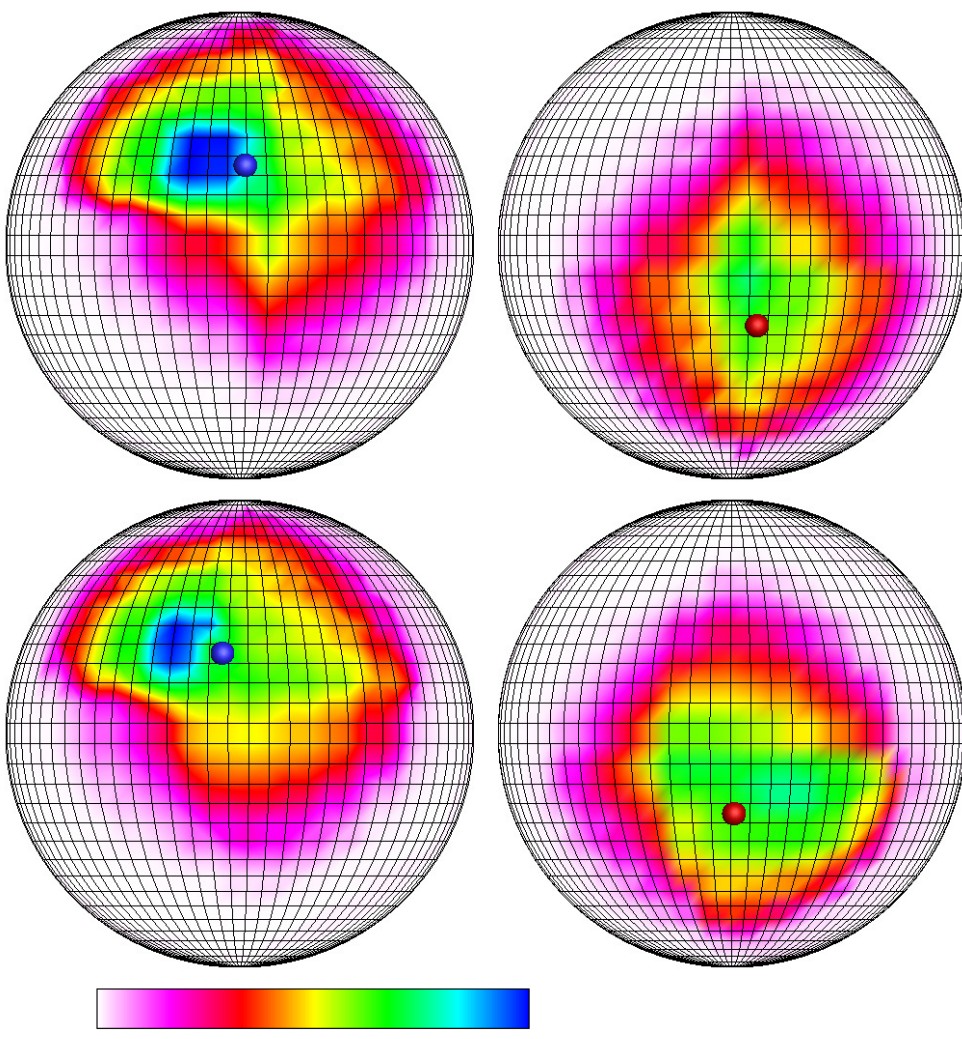

**Figure 16.** Temperature distribution over the accretor surface for the asynchronous polar with an non-offset dipole when switching the flow from the north magnetic pole to the south one. Temperature maps for Phase 1 (**upper** panel) and for Phase 2 (**lower** panel) are shown.

Upon completion of the switching (Figure 15, lower panel), the jet matter falls only in the vicinity of the north magnetic pole, while the area of the hot spot decreases slightly, as well as its temperature. The deviation value of the hot spot from the magnetic pole is also reduced, which now amounts to $15°$. When comparing the accretion rates to the southern spot before the pole switching and to the northern spot after its completion (see Table 4, Phase 1 and Phase 4), it can be concluded that in the last case, the tempo is decreased slightly. A similar pattern is observed with reverse switching (see Table 5, Phase 1 and Phase 4): the accretion rate to the southern spot is 20% higher than to the northern one. This is probably due to a partial loss of the flow velocity during the movement of matter in the vicinity of the north magnetic pole. We also note the fact that after the completion of the first switching, the symmetry of the areas of increased temperature relative to the poles changes: now, their center coincides with the center of the hot spot.

When switching the flow from the north magnetic pole to the south one (Figures 16 and 17), the changes in the parameters of hot spots and their drift occur, according to the scenario just described.

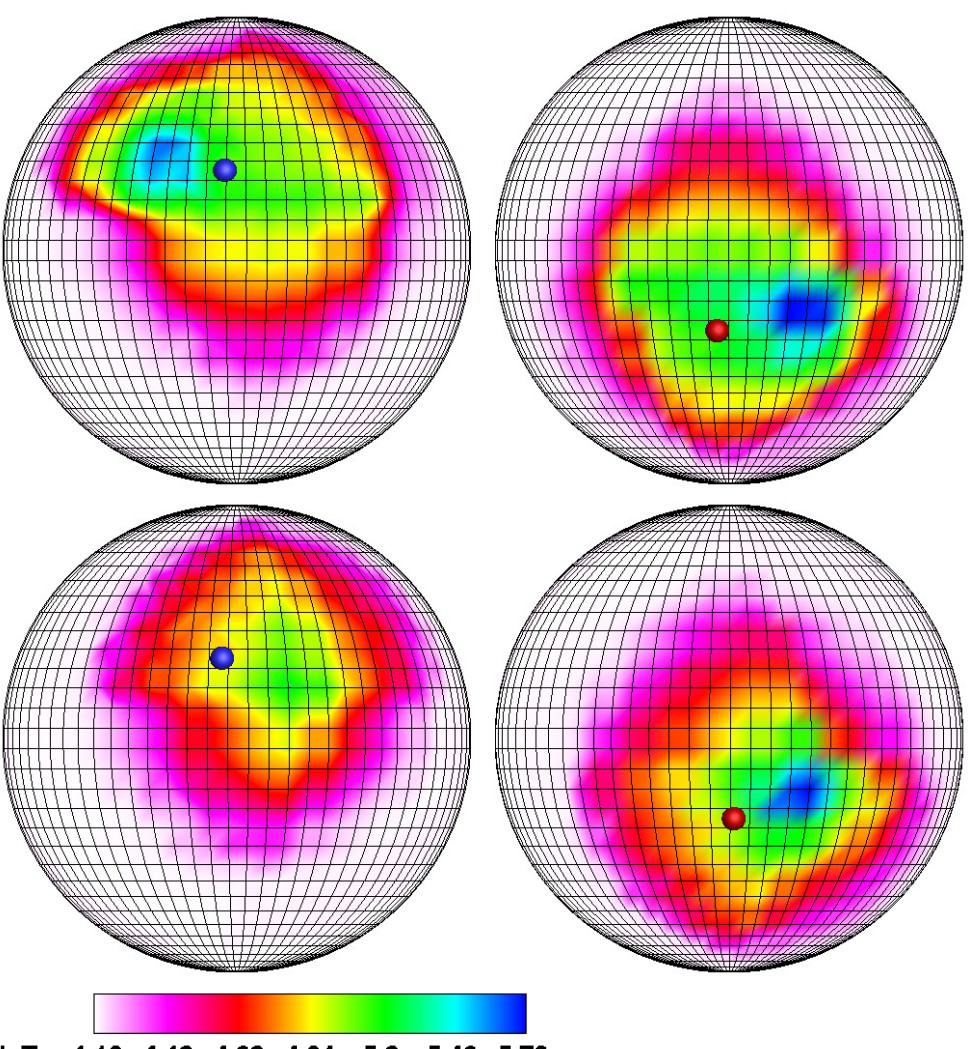

**lgT:** 4.16  4.42  4.68  4.94  5.2  5.46  5.72

**Figure 17.** The same as in Figure 16, but for Phase 3 (**upper** panel) and Phase 4 (**lower** panel).

Figures 18–21 show the temperature distribution over the accretor surface for the case of the offset magnetic dipole.

Note that when the dipole axis deviates from the center of the accretor, the equivalence of the magnetic poles is violated. First of all, this is expressed in the difference in the values of the field strength in the vicinity of each pole. In this case, the accretion rate of the jet matter will depend on two factors: the value of magnetic field induction in vicinity of the pole and the distance from the inner Lagrange point to pole (the orientation of dipole axis relative to this point). Keeping this in mind, the configuration of the offset dipole considered in this paper can be characterized as follows. Since the axis of the dipole is shifted below the center of the white dwarf by half the radius of the star, it is obvious that the north magnetic pole will have a stronger influence on the behavior of the jet than the south one. This is due to the fact that with the proper asynchronous rotation of the accretor, the north pole will pass closer to the inner Lagrange point compared to the south pole. On the other hand, the magnetic field strength near the south pole is noticeably greater than that of the north one. In this regard, when assessing the drift of hot spots, we should expect a smaller deviation of the northern accretion zone, compared to the southern one. However, the constructed temperature maps for the offset dipole indicate the opposite. Thus, when switching the flow from south pole to the north one (Figures 18 and 19), it can be seen that at the initial phase of the process, the southern spot is very tightly adjacent to the corresponding pole, and when the energy release zone occurs in the vicinity of the

north magnetic pole at the final stages of the process, the latter is as far away from the pole as possible. When the flow is switched back from the north pole to the south one (Figures 20 and 21), an observed picture is more consistent with the theoretical reasoning: at the initial stage, the northern spot is located close to the pole, and at the end, the southern spot is removed at a considerable distance from the pole. However, the described location of the spots is in good agreement with the flow geometry shown in Figures 10–13.

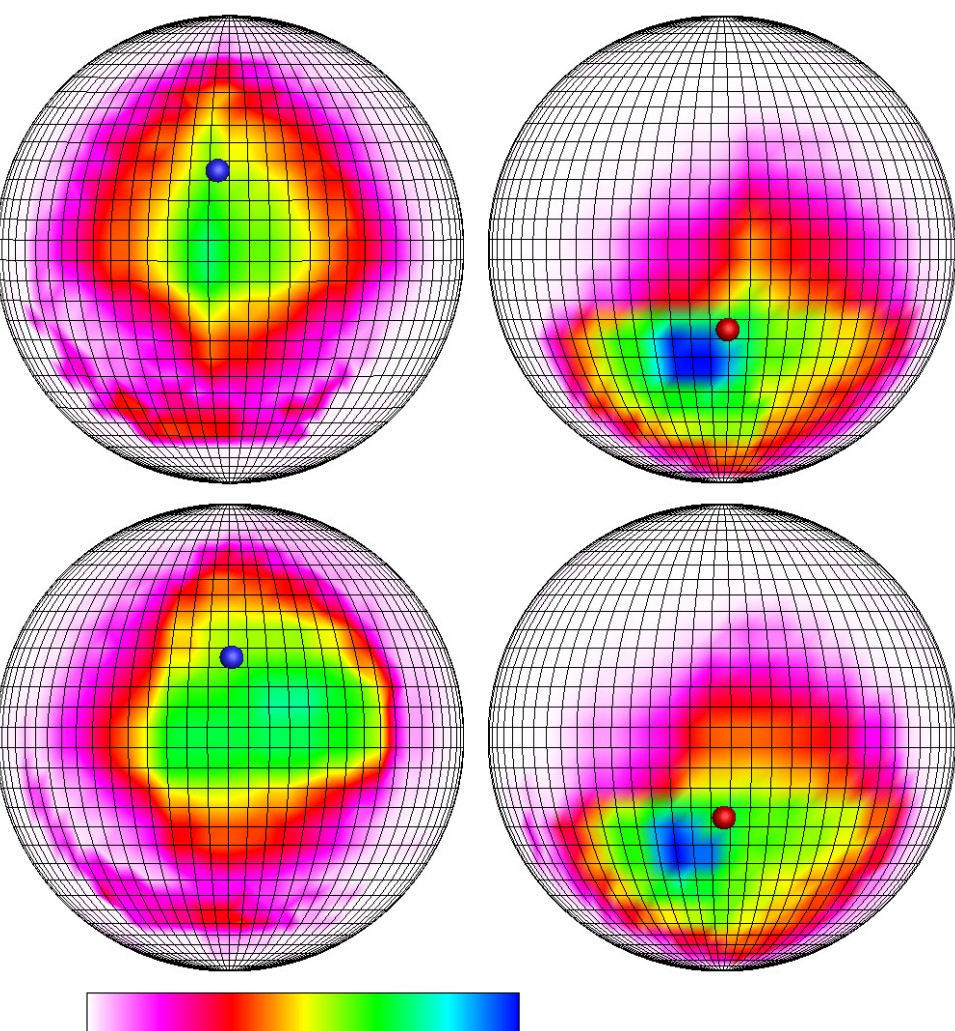

**Figure 18.** Temperature distribution over the accretor surface for the asynchronous polar with an offset dipole when switching the flow from the south magnetic pole to the north one. The temperature maps for Phase 1 (**upper** panel) and for Phase 2 (**lower** panel) are shown.

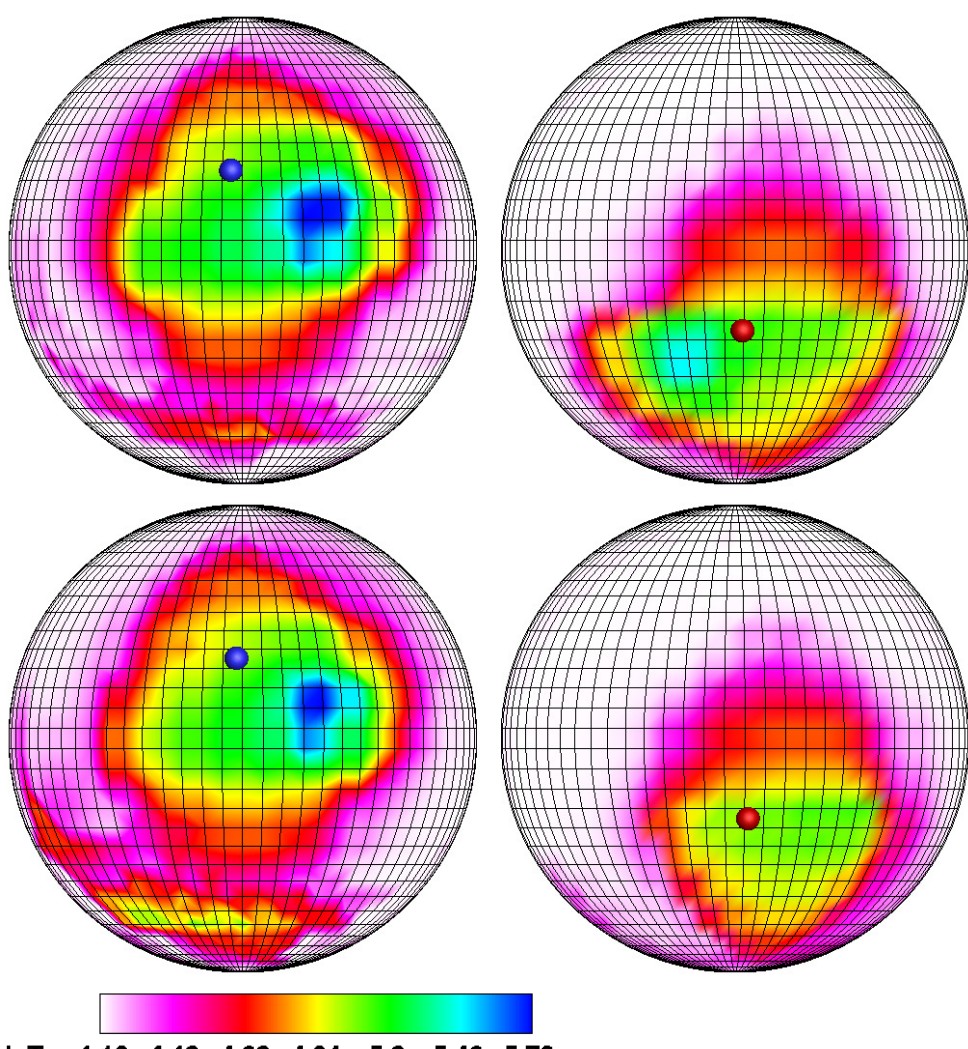

**Figure 19.** The same as in Figure 18, but for Phase 3 (**upper** panel) and Phase 4 (**lower** panel).

At the first switching, due to the considerable length of the ballistic part of the trajectory and taking into account the fact that the axis of the dipole is perpendicular to the line connecting the donor and the accretor centers, the southern spot at the initial moment of time (Figure 18, upper panel) is located close to the pole, and its center has a deviation of about 5°. At the same time, at the formation moment of the arch of the matter (Figure 18, lower panel), the southern spot retains its position, having lost only a little in area, and the weakly pronounced northern spot is already 10° away from the pole at this stage. Then, at the phase of maximum convergence of hot spots (Figure 19, upper panel), the angles of their deviation are 15° for the south and 20° for the north. Note here that the size of the southern spot still does not change, while the northern spot is slightly elongated along the latitude of the star. Finally, after the switching is completed (Figure 19, lower panel), the accretion of matter from the common envelope occurs in the vicinity of the south pole, and the northern hot spot has reduced its size by two times in longitude and has retained the magnitude of deviation from the pole.

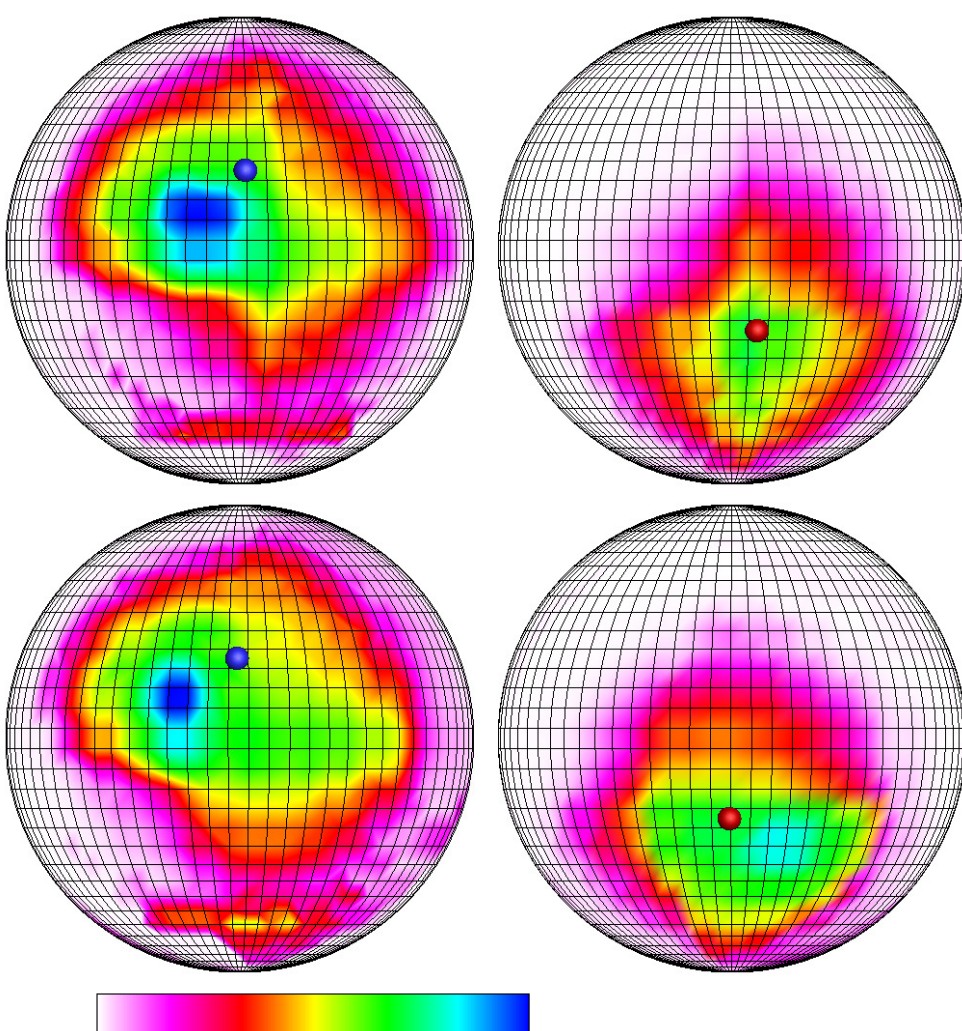

**Figure 20.** Temperature distribution over the accretor surface for the asynchronous polar with an offset dipole when switching the flow from the north magnetic pole to the south one. The temperature maps for Phase 1 (**upper** panel) and for Phase 2 (**lower** panel) are shown.

When switching the flow from the north magnetic pole to the south one (Figures 20 and 21), an opposite picture can be observed. At the initial stage (Figure 20, upper panel), the northern spot is somewhat closer to the pole, but is not located tightly to it, unlike the southern spot at the same phase in the previous switching. Now its deviation is about 15° and besides, it turns out to be shifted in latitude by an angle of 10°. At the phase of arch formation (Figure 20, lower panel), the area of the northern spot is halved, but the position remains unchanged. The southern spot at this time is formed at a distance of 10° from the pole. The magnitude of spot deviation in longitude from the poles at the stage of maximum convergence (Figure 21, upper panel) turns out to be approximately equal and amounted to 20° for each energy release zone. The southern zone is also moved away from the pole in latitude by 10°. At the final stage of the process (Figure 21, lower panel), it can be seen that the southern spot retains its size and position relative to the pole, and the accretion region near the north pole, which is currently receiving matter from the polar common envelope, is shifted 20° further south in latitude.

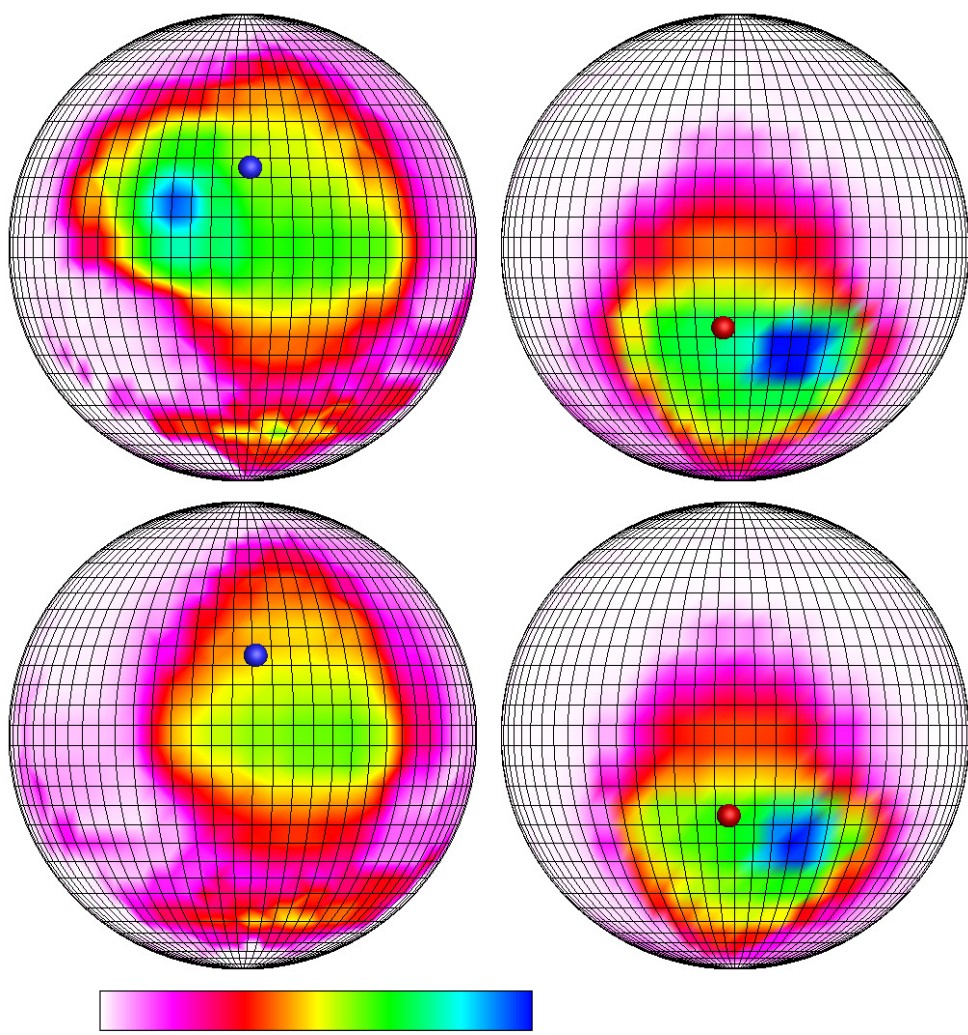

**IgT:** 4.16  4.42  4.68  4.94  5.2  5.46  5.72

**Figure 21.** The same as in Figure 20, but for Phase 3 (**upper** panel) and Phase 4 (**lower** panel).

## 4. Conclusions

In this paper, parameters of hot spots movement on the accretor surface are investigated for two types of polars—synchronous and asynchronous. During the calculations, it was assumed that a prototype of a synchronous polar is the eclipsed V808 Aur system, and that of a non-eclipsed asynchronous polar is the CD Ind system.

A change in the ratio of ballistic and magnetic parts of the jet trajectory is considered a factor affecting the drift of hot spots. For the synchronous polar, in assuming the configuration of the magnetic field with a non-offset dipole, the position of a hot spot is determined by the mass transfer rate. For the asynchronous polar, if it also has a non-offset dipole field configuration, when the value of mass transfer rate is fixed, the rotation of dipole axis over time begins to play an essential role in the drift of spots—a change in its orientation during the beat period relative to the donor.

When studying the flow structure in the typical synchronous polar V808 Aur, four variants of three-dimensional numerical calculations were performed within the stationary model, corresponding to the different states of system and the value of mass transfer rate—from $\dot{M} = 10^{-10}\ M_{\odot}/\text{year}$ (low state) to $\dot{M} = 10^{-7}\ M_{\odot}/\text{year}$ (high state). Calculations have shown that with an increase in the mass transfer rate, the length of ballistic part of the accretion jet grows, where the flow of matter is controlled mainly by gravity and inertia forces (centrifugal force and Coriolis force). At the high state, the length of the ballistic part of the jet trajectory turns out to be maximal, and therefore, the flow has time

to deviate significantly from the direction to the accretor due to the action of the Coriolis force. At the low state, the jet matter, having barely left the inner Lagrange point, almost immediately begins to be controlled by the magnetic field of the accretor, and the ballistic part of trajectory is practically absent. With the chosen geometry of the magnetic field (the north pole is directed to the inner Lagrange point), accretion goes mainly to the north magnetic pole, and the southern accretion zone is much less intense since it is formed due to the flow of matter from the common envelope. The shape, intensity, and location of the northern hot spot significantly depend on the accretion rate. At the high state, it is shifted in longitude relative to the north magnetic pole by about 30°, and in latitude by 5°–7°. At the same time, its area is maximal. When the accretion rate drops by 10 times, the spot continues to drift only in longitude, its deviation from the pole decreases to 15°, but the area of the energy release zone is reduced by two times. At the middle state of the polar, the spot, while maintaining its geometric dimensions, approaches the magnetic pole closely. At the same time, it shifts in longitude and latitude to 5°. Finally, at the low state, the northern energy release zone practically coincides with the north magnetic pole, and its area is 25% of the initial value at the high state.

The investigation of the flow structure in the asynchronous polar CD Ind was performed for two variants of magnetic field configuration—a non-offset and an offset dipole. At the same time, a fixed mass transfer rate was set equal to $\dot{M} = 10^{-9} \, M_\odot / \text{year}$. It should be noted that in asynchronous polars, changing the position of the magnetic pole is a slow process, so it seems appropriate to distinguish four stages of accretion. The first two of them are associated with the fallout of jet matter on one of the magnetic poles and each lasts for about 40 orbital periods of the system. At these stages, the motion of the hot spot can be described analytically, and its displacement from the magnetic pole is determined by the processes discussed above for the synchronous polar. At the other two stages, when the jet switches between the magnetic poles, there is a simultaneous change in both the ballistic and magnetic parts of the jet, which leads to a complex picture of the hot spots movement.

To illustrate the drift of hot spots at the pole switching stages, the calculation results were presented for four characteristic time points: the initial phase, on which the jet matter accretes only to one pole, the phase of the beginning of the jet closure to the opposite pole, accompanied by the formation of an arch of matter in the magnetosphere of the accretor, the phase of maximum mutual convergence of hot spots with a decrease in the inner radius of the arch, and the final phase, when the jet matter is completely switched to the other magnetic pole.

The obtained results of the calculations for the asynchronous polar with a non-offset dipole allow us to conclude that the values of the hot spots' deviation from the magnetic poles, determined by the processes of switching the flow between the poles, are comparable to the displacements of spots observed in the synchronous polar with variations in the mass transfer rate. However, the parameters of spots in the asynchronous polar have a number of features. When switching from the south pole to the north one in the initial phase, a hot spot is tightly adjacent to the pole, but at the same time, it is elongated in longitude and has a maximum area. For comparison, in the synchronous polar at the middle state, a similar spot is half as small. This indicates that during asynchronous rotation and placement of the magnetic pole close to the equator of the primary component, the hot spot stretches along the longitude in accordance with the distribution of flow density in the jet. At the formation moment of an arc of matter, the deviation of both spots from the magnetic poles does not exceed 15° in longitude. At the same time, the area of the southern spot decreases by two times, and the northern spot has a weakly pronounced character. At the maximum convergence of the hot spots on the third switching phase, the deviation angle of both spots from the magnetic poles increases and is approximately 20°. Due to the different positions of the magnetic poles relative to the inner Lagrange point, the temperature of the southern hot spot is noticeably lower than the northern one at this stage. Upon completion of the switching, the jet matter falls only in the vicinity of the north magnetic pole, while the area of the hot spot decreases slightly, as well as its temperature. The magnitude of the

hot spot deviation from the magnetic pole is also reduced, which now amounts to 15° in longitude. Note that the displacement of spots along the latitude in the considered flow configuration is insignificant and does not exceed 5°. The reverse switching of flow from the north magnetic pole to the south one occurs according to the same scenario, and the characteristics of the hot spots' drift are almost identical.

The displacement of the magnetic dipole relative to the center of the accretor leads to an equivalence violation of the magnetic poles, and, accordingly, to an inequality of the accretion parameters of the jet matter. In the field configuration under consideration, the north pole is stronger than the south one since it is located closer to the equator of the white dwarf. In this regard, when assessing the drift of the hot spots, we should expect a smaller deviation of the northern accretion zone, compared to the southern one. However, the constructed temperature maps for the offset dipole indicate that this state is partially fulfilled. When switching the flow from the south pole to the north one, it can be seen that at the initial stage, the southern spot is very tightly adjacent to the corresponding pole, and its center has a deviation of about 5°. At the formation moment of the arch of matter, this spot retains its position, having lost only a little in the area, and the weakly pronounced northern spot is already 10° away from the pole at this stage. Further, at the phase of maximum convergence of the hot spots, the angles of their deviation are 15° for the southern and 20° for the northern, while the size of the southern spot still does not change, and the northern spot is slightly elongated along the latitude of star. Finally, at the end of the switching process, the northern hot spot reduces its size by two times in longitude, along with that maintaining the maximum deviation from the pole.

When switching the flow from the north magnetic pole to the south one, an opposite picture is observed. At the initial stage, the northern spot is somewhat closer to the pole, but is not located tightly to it, unlike the southern spot at the same phase in the previous switching. Now its deviation is about 15° and besides, it turns out to be shifted in latitude by an angle of 10°. At the phase of arch formation, the area of the northern spot is halved, and the position remains unchangeable. The southern spot at this time is formed at a distance of 10° from the pole. At the stage of maximum convergence, the magnitude of the spots' deviation in longitude from the poles turns out to be approximately equal and amount to 20° for each energy release zone. The southern zone is also moved away from the pole in latitude by 10°. At the final stage of the process, the southern spot retains its size and position relative to the pole, and an accretion region near the north pole, which currently receives matter from the polar common envelope, is shifted 20° further south in latitude.

The analysis of the calculated hot spot movements for various types of polars suggests that the deviations of spots from the magnetic poles can be significant—up to 15° in latitude and 30° in longitude. The revealed features of the spot drift provide methodological grounds for determining the most important parameters of the system: for synchronous polars, it is possible to conduct an independent assessment of the mass transfer rate, and for asynchronous polars, to determine the configuration of the magnetic field of the accretor.

**Author Contributions:** D.B. supervised the project, developed a concept of the manuscript, make edits. A.S. prepared the manuscript and performed all the numerical simulations used in this work. A.Z. developed the using numerical code, make edits. All authors have read and agreed to the published version of the manuscript.

**Funding:** The work was supported by the Russian Foundation for Basic Research (project 19-52-60001).

**Institutional Review Board Statement:** Not applicable.

**Informed Consent Statement:** Not applicable.

**Data Availability Statement:** The data presented in this study are available on request from the corresponding author.

**Acknowledgments:** The work was performed using the equipment of the center for collective use "Complex of modeling and data processing of mega-class research facilities" SIC "Kurchatov Insti-

tute", http://ckp.nrcki.ru/ (accessed from May 2019 to November 2020). The work was performed using the computing cluster of the Interdepartmental Supercomputer Center of the Russian Academy of Sciences.

**Conflicts of Interest:** The authors declare no conflict of interest.

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
