# Peer review of "Hot Spots Drift in Synchronous and Asynchronous Polars: Results of Three-Dimensional Numerical Simulation"

_galaxies, doi:10.3390/galaxies9040110_

Round 1
Reviewer 1 Report
The suggestions in my first report would have benefitted the reader.
Author Response
Dear reviewer,
We are sending a corrected version of our paper. I would like to thank you for the comments. We have taken most of them into account in the presented version of the paper. The corrections made are marked in red.
Q1. In model description, lines 134 onwards: I am not convinced that the detailed mathematical description of the model belongs in this manuscript. Most model details have been introduced elsewhere and could easily be referenced; I believe that the reader would benefit much more from a synopsis, i.e. a description of the premises and assumptions, and move the description of the model into the appendix or remove it from the paper entirely, providing references to previous descriptions of the model. I am fairly conversant in the MHD formalism and do not shy away from complex mathematical descriptions, but I do not see a purpose of such a description here -- it attempts to provide details on the computation but it is not comprehensive enough to provide the full picture.
A1. Indeed, the description of the mathematical model presented in the paper is only a part of its full version, given in our previous works. This part describes the features of the numerical method we used for this problem. We would like to leave this description unchanged as one of the sections of the paper, but if you still insist, we can put a more detailed description of the model in the appendix.
Q2. Most 3D figures of accretion flow in the paper have very limited information content. They certainly have a wow factor, but not much beyond that. I would leave one or two, but otherwise try to come up with a better visualization that can quantify the effects better. For those that do stay, a color bar would be much appreciated. Depiction of a purely ballistic flow and a purely magnetic flow might be most useful as the reader would then be able to "interpolate" between the two based on the explored parameters (mass transfer rate, magnetic field strength, synchronicity, etc.). Figures of surface temperature distribution are significantly more telling. I fully appreciate that these are time-expensive calculations and it is not straight-forward to do a proper sweep across parameters, but at least for the combinations already explored, it seems to me that density histograms would fare better at conveying the information to the reader.
A2. The main purpose of these 3D images is to show qualitative changes in the flow pattern of matter during mass exchange between components, the dramatic nature of the change in the flow structure in the case of switching magnetic poles for an asynchronous polar. From these figures, the reader should understand how the flow pattern in each particular case affects the drift of hot spots, which should be expected on the obtained maps of the temperature distribution over the surface of the white dwarf. At the same time, each temperature map corresponds to its own flow pattern, so it is impossible to leave only a few 3D images of the flow structure. Also, these images were not intended to display any flow parameters quantitatively, in particular the length of the ballistic or magnetic part of the jet trajectory. Quantitative analysis of the ratio between these lengths is not part of the objectives of our paper, we only assert that this ratio affects the magnitude of the deviation of the hot spot from the magnetic pole of the accretor, and the figures clearly show that one part of the jet trajectory is larger or smaller than the other. The text of the paper also contains relevant explanations for each figure. We agree that it is necessary to place a density scale in these figures for clarity, this will be done in the corrected version of the paper.
Q3. It is unclear how the surface of the white dwarf is treated for conditioning the MHD flow. Looking at Fig. 1, for example, there are small increased density regions on the opposite side of the magnetic axis, where I would not expect a large contribution from the common envelope. Is that an artifact or a real phenomenon? Also, the text mentions that accretion to the second pole is caused by the flow from the common envelope through L2, which I find confusing: any mass that goes past L2 will be irretrievably lost, so the only inflow that is plausible would be from matter within the L2 lobe. It would be good to elucidate these points.
A3. In our numerical calculations, there is one of the boundary conditions - the density at the boundary of the computational domain, which is small, but not equal to zero. It is based on the fact that even in polars there is some rarified matter between components in the system. This matter could be considered as a common envelope in the binary system, the formation of which occurs by two mechanisms: due to part of the stream from the inner Lagrange point L1, which does not include the main mass transfer flow, but dissipates in the system, and due to accreting of the matter flowing from the outer point L2 under action of dissipative processes. We tried to clarify this in the paper.
Q4. What is the justification for the 1/2 radius offset of the magnetic axis? I understand that it is just a qualitative proof-of-concept, but does the offset amount have any physical significance?
A4. The polar model with 1/2 radius offset dipole is based on the observational data of the real system CD Ind (there is a link to this work in the paper: P. Hakala, G. Ramsay, S. B. Potter, A. Beardmore, D. H. Buckley, and G. Wynn, Monthly Not. Roy. Astron. Soc. 486, 2549 (2019)). We understand that the real magnetic field could be more complex, but as it is obvious for observers the real field can be described by some effective model of an offset dipole. We plan to study the influence of the offset onto the polar flow structure in a separate paper.Q5. The paper can be shortened by ~20-30% without loss of content -- there are many points that are repeated and the figures - while generally instructive - could be further distilled.
A5. In the corrected version of the paper, we will try to revise the content and somewhat reduce its volume.
MINOR POINTS:
Q6. The level of English is quite good, but grammar construct edits are necessary. This is the job for the editorial team.
A6. It has done.
Q7. Section 1 is light on citations -- beyond the 1st paragraph, most of the section introduces concepts but with no references. The ones that are given are, for the most part, self-citations. It would be welcome to get a sense of what other groups around the world have done. There is a fair number of statements that are presented ad-hoc, i.e. without a reference and without being established. For example, in the Introduction, line 52, "In addition, the deflection of the jet is facilitated by action of the Coriolis force." This is undoubtedly the case, but how so, to what extent, and how relevant is its contribution compared to ballistic/magnetic components is not discussed.
A7. The text of the paper has been corrected taking into account your comment, references to the facts and statements from the works of other authors have been added.
Q8. The discussion of reference frames in Section 2 warrants a figure; reading the description 108-123 is quite tedious because of a non-standard setup of the coordinate system. I ended up having to draw everything myself just to make sure I understand, and I had to read the text several times to catch all the nuances correctly. I suspect that other readers would thoroughly appreciate a helper diagram.
A8. Explanatory diagram added to the paper.
Q9. Line 124: I never heard of the field being referred to as "potential" when curl is 0; do the authors mean irrotational?
A9. Yes, it means irrotational. It was corrected. In russian literature field with no curl called potential, so it`s a translate error.
Q10. Section 3, lines 174-186: the authors should consider tabulating this information.
A10. Tables was added to the paper.
Q11. Line 214: it -> we?
A11. Yes, we. But it possible to write this part of sentence like this with no change in sence: it can be seen all the structure...
Q12. Line 235: what is the temperature of energy release zone? Is this the hot spot? The authors should clarify what they mean.
A12. In the given case the temperature as indicated is «in oder 106 K». A hot spot and an energy release zone are the same terms. Both are used in the text of the paper to avoid repetition.
Q13. Lines 271-272: where does the ~40 orbital periods estimate come from? It sounds completely ad-hoc to me.
A13. Based on the initial data and using formula (1), it can be calculated that approximately 100 orbital periods of the system correspond to one beat period of the asynchronous polar. Of these, about 20 account for switching processes, the remaining 80 for two stages of unipolar accretion of 40 periods each.
Q14. Line 274: "can be described analytically" -- this is an assumption, not a fact -- the authors chose to parametrize the change analytically.
A14. Yes, it`s assumption.
Q15. Lines 282-284: is it a safe assumption that, if the magnetic axis is shifted, the field remains dipole in structure? Thinking of the dynamo process that creates the offset field, I would expect it to likely not be (exclusively) dipolar.
A15. This assumption is model initial condition based on estimates of observers (see A.4)
Q16. Lines 285-297: this would be easier to parse if it was tabulated.
A16. It has done
Q17. Figs. 5-8 could probably be combined into a single figure.
A17. Each figure of the flow structure includes two panels for the convenience of pointing to them in the text. In addition, combining Fig. 5-8 into one will significantly reduce the scale of each panel, which will cause it to lose in detail. We think this is unreasonable.
Q18. I quite enjoyed the interpretative part of the paper; it is a little long-winded but it does convey the central part of the paper. I would only suggest stressing that these are inherently qualitative conclusions, where robustness of these inherently chaotic flows cannot really be assessed in a straight-forward manner (i.e., sensitivity to parameter variations).
A18. In this peper, we limited ourselves to a qualitative description of the flow structure in the polars, since the main task was to study the drift of hot spots. However, a quantitative assessment of the parameters of the flow itself is possible based on the results of numerical calculations.
In conclusion, I remain enthusiastic about the group's MHD work and appreciate the results disseminated in the paper. I would love to see that the authors consider the raised points and address them as they see suitable.
Reviewer 2 Report
Review of the manuscript titled "Hot Spots Drift in Synchronous and Asynchronous Polars: Results of Three-Dimensional Numerical Simulation" by Dmitry Bisikalo, Andrey Sobolev and Andrey Zhilkin
Nov 1, 2021
The manuscript presents the results of the magneto-hydrodynamical (MHD) simulations of accretion flow in polarized cataclysmic variables (polars). These objects are known for their pronounced magnetic fields that influence the flow of accreted matter from the donor (typically a main sequence star) to the accretor (typically a white dwarf). The authors focus on the interplay between ballistic and magnetic accretion in two types of polars: the ones with a constant magnetic axis that goes through the center of the accretor (synchronous) and the other where the magnetic axis changes and is either centered or offset (asynchronous). In addition, the authors establish that the location and temperature of the donor-facing hot spot on the accretor depends on mass transfer rate while the hot spot on the opposite side does not, because it accretes matter from the common envelope rather than the stream through L1. The paper is mostly qualitative in context, insightful and of immediate interest to the binary star community; below I provide my comments and concerns in the hope that they help the authors with an independent review.
MAJOR POINTS:
* In model description, lines 134 onwards: I am not convinced that the detailed mathematical description of the model belongs in this manuscript. Most model details have been introduced elsewhere and could easily be referenced; I believe that the reader would benefit much more from a synopsis, i.e. a description of the premises and assumptions, and move the description of the model into the appendix or remove it from the paper entirely, providing references to previous descriptions of the model. I am fairly conversant in the MHD formalism and do not shy away from complex mathematical descriptions, but I do not see a purpose of such a description here -- it attempts to provide details on the computation but it is not comprehensive enough to provide the full picture.
* Most 3D figures of accretion flow in the paper have very limited information content. They certainly have a wow factor, but not much beyond that. I would leave one or two, but otherwise try to come up with a better visualization that can quantify the effects better. For those that do stay, a color bar would be much appreciated. Depiction of a purely ballistic flow and a purely magnetic flow might be most useful as the reader would then be able to "interpolate" between the two based on the explored parameters (mass transfer rate, magnetic field strength, synchronicity, etc.). Figures of surface temperature distribution are significantly more telling. I fully appreciate that these are time-expensive calculations and it is not straight-forward to do a proper sweep across parameters, but at least for the combinations already explored, it seems to me that density histograms would fare better at conveying the information to the reader.
* It is unclear how the surface of the white dwarf is treated for conditioning the MHD flow. Looking at Fig. 1, for example, there are small increased density regions on the opposite side of the magnetic axis, where I would not expect a large contribution from the common envelope. Is that an artifact or a real phenomenon? Also, the text mentions that accretion to the second pole is caused by the flow from the common envelope through L2, which I find confusing: any mass that goes past L2 will be irretrievably lost, so the only inflow that is plausible would be from matter within the L2 lobe. It would be good to elucidate these points.
* What is the justification for the 1/2 radius offset of the magnetic axis? I understand that it is just a qualitative proof-of-concept, but does the offset amount have any physical significance?
* The paper can be shortened by ~20-30% without loss of content -- there are many points that are repeated and the figures -- while generally instructive -- could be further distilled.
MINOR POINTS:
* The level of English is quite good, but grammar construct edits are necessary. This is the job for the editorial team.
* Section 1 is light on citations -- beyond the 1st paragraph, most of the section introduces concepts but with no references. The ones that are given are, for the most part, self-citations. It would be welcome to get a sense of what other groups around the world have done. There is a fair number of statements that are presented ad-hoc, i.e. without a reference and without being established. For example, in the Introduction, line 52, "In addition, the deflection of the jet is facilitated by action of the Coriolis force." This is undoubtedly the case, but how so, to what extent, and how relevant is its contribution compared to ballistic/magnetic components is not discussed.
* The discussion of reference frames in Section 2 warrants a figure; reading the description 108-123 is quite tedious because of a non-standard setup of the coordinate system. I ended up having to draw everything myself just to make sure I understand, and I had to read the text several times to catch all the nuances correctly. I suspect that other readers would thoroughly appreciate a helper diagram.
* Line 124: I never heard of the field being referred to as "potential" when curl is 0; do the authors mean irrotational?
* Section 3, lines 174-186: the authors should consider tabulating this information.
* Line 214: it -> we?
* Line 235: what is the temperature of energy release zone? Is this the hot spot? The authors should clarify what they mean.
* Lines 271-272: where does the ~40 orbital periods estimate come from? It sounds completely ad-hoc to me.
* Line 274: "can be described analytically" -- this is an assumption, not a fact -- the authors chose to parametrize the change analytically.
* Lines 282-284: is it a safe assumption that, if the magnetic axis is shifted, the field remains dipole in structure? Thinking of the dynamo process that creates the offset field, I would expect it to likely not be (exclusively) dipolar.
* Lines 285-297: this would be easier to parse if it was tabulated.
* Figs. 5-8 could probably be combined into a single figure.
* I quite enjoyed the interpretative part of the paper; it is a little long-winded but it does convey the central part of the paper. I would only suggest stressing that these are inherently qualitative conclusions, where robustness of these inherently chaotic flows cannot really be assessed in a straight-forward manner (i.e., sensitivity to parameter variations).
In conclusion, I remain enthusiastic about the group's MHD work and appreciate the results disseminated in the paper. I would love to see that the authors consider the raised points and address them as they see suitable.
Author Response
Dear reviewer,
We are sending a corrected version of our paper. I would like to thank you for the comments. We have taken most of them into account in the presented version of the paper. The corrections made are marked in red.
Q1. In model description, lines 134 onwards: I am not convinced that the detailed mathematical description of the model belongs in this manuscript. Most model details have been introduced elsewhere and could easily be referenced; I believe that the reader would benefit much more from a synopsis, i.e. a description of the premises and assumptions, and move the description of the model into the appendix or remove it from the paper entirely, providing references to previous descriptions of the model. I am fairly conversant in the MHD formalism and do not shy away from complex mathematical descriptions, but I do not see a purpose of such a description here -- it attempts to provide details on the computation but it is not comprehensive enough to provide the full picture.
A1. Indeed, the description of the mathematical model presented in the paper is only a part of its full version, given in our previous works. This part describes the features of the numerical method we used for this problem. We would like to leave this description unchanged as one of the sections of the paper, but if you still insist, we can put a more detailed description of the model in the appendix.
Q2. Most 3D figures of accretion flow in the paper have very limited information content. They certainly have a wow factor, but not much beyond that. I would leave one or two, but otherwise try to come up with a better visualization that can quantify the effects better. For those that do stay, a color bar would be much appreciated. Depiction of a purely ballistic flow and a purely magnetic flow might be most useful as the reader would then be able to "interpolate" between the two based on the explored parameters (mass transfer rate, magnetic field strength, synchronicity, etc.). Figures of surface temperature distribution are significantly more telling. I fully appreciate that these are time-expensive calculations and it is not straight-forward to do a proper sweep across parameters, but at least for the combinations already explored, it seems to me that density histograms would fare better at conveying the information to the reader.
A2. The main purpose of these 3D images is to show qualitative changes in the flow pattern of matter during mass exchange between components, the dramatic nature of the change in the flow structure in the case of switching magnetic poles for an asynchronous polar. From these figures, the reader should understand how the flow pattern in each particular case affects the drift of hot spots, which should be expected on the obtained maps of the temperature distribution over the surface of the white dwarf. At the same time, each temperature map corresponds to its own flow pattern, so it is impossible to leave only a few 3D images of the flow structure. Also, these images were not intended to display any flow parameters quantitatively, in particular the length of the ballistic or magnetic part of the jet trajectory. Quantitative analysis of the ratio between these lengths is not part of the objectives of our paper, we only assert that this ratio affects the magnitude of the deviation of the hot spot from the magnetic pole of the accretor, and the figures clearly show that one part of the jet trajectory is larger or smaller than the other. The text of the paper also contains relevant explanations for each figure. We agree that it is necessary to place a density scale in these figures for clarity, this will be done in the corrected version of the paper.
Q3. It is unclear how the surface of the white dwarf is treated for conditioning the MHD flow. Looking at Fig. 1, for example, there are small increased density regions on the opposite side of the magnetic axis, where I would not expect a large contribution from the common envelope. Is that an artifact or a real phenomenon? Also, the text mentions that accretion to the second pole is caused by the flow from the common envelope through L2, which I find confusing: any mass that goes past L2 will be irretrievably lost, so the only inflow that is plausible would be from matter within the L2 lobe. It would be good to elucidate these points.
A3. In our numerical calculations, there is one of the boundary conditions - the density at the boundary of the computational domain, which is small, but not equal to zero. It is based on the fact that even in polars there is some rarified matter between components in the system. This matter could be considered as a common envelope in the binary system, the formation of which occurs by two mechanisms: due to part of the stream from the inner Lagrange point L1, which does not include the main mass transfer flow, but dissipates in the system, and due to accreting of the matter flowing from the outer point L2 under action of dissipative processes. We tried to clarify this in the paper.
Q4. What is the justification for the 1/2 radius offset of the magnetic axis? I understand that it is just a qualitative proof-of-concept, but does the offset amount have any physical significance?
A4. The polar model with 1/2 radius offset dipole is based on the observational data of the real system CD Ind (there is a link to this work in the paper: P. Hakala, G. Ramsay, S. B. Potter, A. Beardmore, D. H. Buckley, and G. Wynn, Monthly Not. Roy. Astron. Soc. 486, 2549 (2019)). We understand that the real magnetic field could be more complex, but as it is obvious for observers the real field can be described by some effective model of an offset dipole. We plan to study the influence of the offset onto the polar flow structure in a separate paper.Q5. The paper can be shortened by ~20-30% without loss of content -- there are many points that are repeated and the figures - while generally instructive - could be further distilled.
A5. In the corrected version of the paper, we will try to revise the content and somewhat reduce its volume.
MINOR POINTS:
Q6. The level of English is quite good, but grammar construct edits are necessary. This is the job for the editorial team.
A6. It has done.
Q7. Section 1 is light on citations -- beyond the 1st paragraph, most of the section introduces concepts but with no references. The ones that are given are, for the most part, self-citations. It would be welcome to get a sense of what other groups around the world have done. There is a fair number of statements that are presented ad-hoc, i.e. without a reference and without being established. For example, in the Introduction, line 52, "In addition, the deflection of the jet is facilitated by action of the Coriolis force." This is undoubtedly the case, but how so, to what extent, and how relevant is its contribution compared to ballistic/magnetic components is not discussed.
A7. The text of the paper has been corrected taking into account your comment, references to the facts and statements from the works of other authors have been added.
Q8. The discussion of reference frames in Section 2 warrants a figure; reading the description 108-123 is quite tedious because of a non-standard setup of the coordinate system. I ended up having to draw everything myself just to make sure I understand, and I had to read the text several times to catch all the nuances correctly. I suspect that other readers would thoroughly appreciate a helper diagram.
A8. Explanatory diagram added to the paper.
Q9. Line 124: I never heard of the field being referred to as "potential" when curl is 0; do the authors mean irrotational?
A9. Yes, it means irrotational. It was corrected. In russian literature field with no curl called potential, so it`s a translate error.
Q10. Section 3, lines 174-186: the authors should consider tabulating this information.
A10. Tables was added to the paper.
Q11. Line 214: it -> we?
A11. Yes, we. But it possible to write this part of sentence like this with no change in sence: it can be seen all the structure...
Q12. Line 235: what is the temperature of energy release zone? Is this the hot spot? The authors should clarify what they mean.
A12. In the given case the temperature as indicated is «in oder 106 K». A hot spot and an energy release zone are the same terms. Both are used in the text of the paper to avoid repetition.
Q13. Lines 271-272: where does the ~40 orbital periods estimate come from? It sounds completely ad-hoc to me.
A13. Based on the initial data and using formula (1), it can be calculated that approximately 100 orbital periods of the system correspond to one beat period of the asynchronous polar. Of these, about 20 account for switching processes, the remaining 80 for two stages of unipolar accretion of 40 periods each.
Q14. Line 274: "can be described analytically" -- this is an assumption, not a fact -- the authors chose to parametrize the change analytically.
A14. Yes, it`s assumption.
Q15. Lines 282-284: is it a safe assumption that, if the magnetic axis is shifted, the field remains dipole in structure? Thinking of the dynamo process that creates the offset field, I would expect it to likely not be (exclusively) dipolar.
A15. This assumption is model initial condition based on estimates of observers (see A.4)
Q16. Lines 285-297: this would be easier to parse if it was tabulated.
A16. It has done
Q17. Figs. 5-8 could probably be combined into a single figure.
A17. Each figure of the flow structure includes two panels for the convenience of pointing to them in the text. In addition, combining Fig. 5-8 into one will significantly reduce the scale of each panel, which will cause it to lose in detail. We think this is unreasonable.
Q18. I quite enjoyed the interpretative part of the paper; it is a little long-winded but it does convey the central part of the paper. I would only suggest stressing that these are inherently qualitative conclusions, where robustness of these inherently chaotic flows cannot really be assessed in a straight-forward manner (i.e., sensitivity to parameter variations).
A18. In this peper, we limited ourselves to a qualitative description of the flow structure in the polars, since the main task was to study the drift of hot spots. However, a quantitative assessment of the parameters of the flow itself is possible based on the results of numerical calculations.
In conclusion, I remain enthusiastic about the group's MHD work and appreciate the results disseminated in the paper. I would love to see that the authors consider the raised points and address them as they see suitable.
This manuscript is a resubmission of an earlier submission. The following is a list of the peer review reports and author responses from that submission.
Round 1
Reviewer 1 Report
- In the section beginning with line 125, equations and definitions of parameters are given. Would it be useful to have a diagram of the binary showing the parameters visually? I'm only asking that it be considered.
- I think that in Line 201 red dwarf should be white dwarf.
- There are just a few grammatical changes that need to be made.
- The conclusions are little long for me. I would like to be able to look at the conclusions, around the length of the abstract, and see clearly what the authors want to highlight in this work.
- I don't see a typical discussion section. "How do the results compare to those in the literature?" I didn't get a strong feeling of how to answer that question.